# Variance-reduced normalized zeroth order method for generalized-smooth Non-convex Optimization

## Abstract

The generalized smooth condition, $(L_0, L_1)$-smoothness, has triggered people's interest since it is more realistic in many optimization problems shown by both empirical and theoretical evidence. To solve the generalized smooth optimization, gradient clipping methods are often employed, and have theoretically been shown to be as effective as the traditional gradient-based methods(Chen et al., 2023; Xie et al., 2024). However, whether these methods can be safely extended to zeroth-order case is still unstudied. To answer this important question, we propose a zeroth-order normalized gradient method(ZONSPIDER) for both finite sum and general expectation case, and we prove that we can find $\epsilon$- stationary point of $f(x)$ with optimal decency on $d$ and $\epsilon$, specifically, the complexes are $\mathcal{O}(d\epsilon^{-2}\sqrt{n}\max\{L_0, L_1\})$ in the finite sum case and $\mathcal{O}(d\epsilon^{-3}\max\{\sigma_1^2, \sigma_0^2\}\max\{L_0, L_1\})$ in the general expectation case. To the best of our knowledge, this is the first time that sample complexity bounds are established for a zeroth-order method under generalized smoothness.

## 1 Introduction

In the paper, we consider solving the following stochastic finite-sum optimization problems. $f : \mathbb{R}^d \to \mathbb{R}$

$$\underset{x \in \mathbb{R}^d}{\text{minimize}} \quad f(x) = \frac{1}{n}\sum_{i=1}^{n} f_i(x) \quad \text{(finite sum case)} \tag{1}$$

where $f(x)$ and each $f_i(x)$ are both differentiable and possibly nonconvex functions, which captures the standard empirical risk minimization problems in machine learning. Additionally, when dealing with a substantial or potentially infinite number of data samples, such as in online or streaming scenarios, we consider the following general expectation optimization problem:

$$\underset{x \in \mathbb{R}^d}{\text{minimize}} \quad f(x) \equiv \mathbb{E}[f(x;\xi)] \quad \text{(general expectation case)} \tag{2}$$

where $\xi$ is a random variable following an unknown distribution.In recent years, significant progress has been made in addressing this problem under the $L$-smooth assumption, with numerous studies contributing to this area.Notable examples include stochastic gradient descent (SGD) (Ghadimi and Lan, 2013), and variance reduction methods (Johnson and Zhang, 2013; Fang et al., 2018; Cutkosky and Orabona, 2019) under stochastic smoothness, which have demonstrated faster convergence. Several works have explored the fastest achievable rates in stochastic optimization. (Han et al., 2024; Zhou and Gu, 2019) established a lower bound of $\mathcal{O}(\epsilon^{-2}\sqrt{n} + n)$ for the finite-sum case, while (Arjevani et al., 2022) set a lower bound of $\mathcal{O}(\epsilon^{-3}\sigma + \epsilon^{-2}\sigma^2)$ for the general expectation case. Despite these strong theoretical results, they all rely on the $L$-smooth assumption, which may not hold in critical applications such as LSTM (Zhang et al., 2019). A key observation is that the smoothness parameter $L$ scales with the gradient norm, leading to the introduction of the generalized $(L_0, L_1)$-smooth assumption:

$$\|\nabla f(x) - \nabla f(x')\| \le (L_0 + L_1\|\nabla f(x)\|) \|x - x'\| \tag{3}$$

Since the traditional $L$-smooth assumption is a special case where $L_1 = 0$, solving problems under the more general $(L_0, L_1)$-smooth condition is harder. To address this, (Zhang et al., 2019) introduced

a gradient clipping method that finds an $\epsilon$-stationary point in $\mathcal{O}(\epsilon^{-4})$ iterations, demonstrating that it can be arbitrarily faster than gradient descent (GD) when the problem has poor initialization. In the traditional $L$-smooth case, variance reduction methods achieve an $\mathcal{O}(\epsilon^{-3})$ rate, motivating exploration of similar techniques under the $(L_0, L_1)$-smooth condition. Recent works (Chen et al., 2023; Reisizadeh et al., 2023) achieved this $\mathcal{O}(\epsilon^{-3})$ rate by incorporating SPIDER (Fang et al., 2018). Thus, the $(L_0, L_1)$-smooth case can be as effective as the traditional $L$-smooth assumption with first-order oracle.

On the other hand, these methods require access to the gradient of the objective function, however in some important applications the explicit expressions of gradients of the objective function are expensive or infeasible to obtain, and only function evaluations are accessible. Such as class of applications include black-box adversarial attacks on deep neural networks (DNNs) (Papernot et al., 2017; Chen et al., 2017) and reinforcement learning (Malik et al., 2018; Kumar et al., 2020). Zeroth-order optimization is a fundamental research topic serving as a prototype module for above numerous tasks. However, all of zeroth-order optimization only studied under traditional $L$-smooth assumption. This motivates us to explore zeroth-order optimization methods under $(L_0, L_1)$-smooth case, as mentioned in the previous discussion, SGD can't be directly applied to $(L_0, L_1)$ case, leading to the natural question:

> Can zeroth-order methods solve generalized $(L_0, L_1)$-smooth nonconvex problems as efficiently as solving traditional smooth nonconvex problems? In particular, what convergence rates can be achieved?

This paper answers this question by proposing a zeroth-order normalized gradient method, which can find a stationary point of $f(x)$ with $\mathcal{O}(d\sqrt{n}\epsilon^{-2}\max\{L_1, L_0\})$ in finite sum case and $\mathcal{O}(d\epsilon^{-3}\max\{L_1, L_0\}\max\{\sigma_1, \sigma_0\})$ in expectation case, both enjoy the optimal dependency on $\epsilon$ and $d$, to the best of our knowledge, this is the first time that sample complexity bounds are established for a zeroth-order method under generalized smoothness.

## 1.1 RELATED WORKS

Among the related works, the most relevant to ours are (Reisizadeh et al., 2023; Chen et al., 2023; Ji et al., 2019a). Compared to (Reisizadeh et al., 2023), while we both analyze SPIDER under the $(L_0, L_1)$-smooth setting, we also explore the zeroth-order case. Their complexity includes an $\mathcal{O}(1/L_1)$ term, which makes their analysis inapplicable to the traditional $L_0$-smooth case, as $1/L_1 \to \infty$. In contrast to (Chen et al., 2023), though both analyze SPIDER under $(L_0, L_1)$-smoothness, we further address the zeroth-order and finite-sum cases. Similarly, compared to (Ji et al., 2019a), while we both use a minibatch version of the rand gradient estimator, we extend the analysis to $(L_0, L_1)$-smoothness and have additional analysis of expectation settings. Our contributions can be summarized as follows:

1. Through the combination of normalized SPIDER and two zeroth-order estimator (called coord and rand gradient estimators), we first give analysis of zeroth-order method under $(L_0, L_1)$-smooth and $(\sigma_0, \sigma_1)$-variance settings, the takeaway of our paper is that zeroth-order method $(L_0, L_1)$ can as effective as in $L$-smooth case. Especially, our method requires weaker assumptions to find an $\epsilon$-stationary point of the black-box optimization problems 1 and 2, as shown in Table 1.

2. We give converge analysis of coord and rand gradient estimators in both finite sum and general expectation cases. Moreover, the proposed methods achieve optimal dependence on $\epsilon$ and $d$, $\mathcal{O}(d\epsilon^{-2}\sqrt{n}\max\{L_0, L_1\})$ in finite sum case and $\mathcal{O}(d\epsilon^{-3}\max\{L_1, L_0\}\max\{\sigma_0^2, \sigma_1^2\})$ in expectation case, which means we can use zeroth-order method to solve $(L_0, L_1)$-smooth problem safely, as shown in Table 2.

3. We conduct experiments to give advice on parameters choice in practice and verify the effectiveness of our method.

Table 1: Assumptions comparison of the representative non-convex methods for finding an $\epsilon$- stationary point of $f(x)$. **Bounded Gradient** denotes $\|\nabla f(x)\| \leq C$ for some constant C. **Bounded Estimator Variance** denotes the bounded variance of rand estimator, i.e, $\mathbb{E}\left[\left\|\bar{\nabla}f(x) - \mathbb{E}[\bar{\nabla}f(x)]\right\|^2\right] \leq \sigma^2$, which is a stronger assumption than bounded gradient variance since its variance scale with the dimension $d$.

| Method | Order | Smoothness | Finite Sum | Expectation case | Bounded Gradient | Bounded Estimator Variance |
|---|---|---|---|---|---|---|
| (Kornowski and Shamir, 2024) | $0^{th}$ | $L$-Lipschitz | ✗ | ✓ | no need | no need |
| (Reisizadeh et al., 2023) | $1^{st}$ | $(L_0, L_1)$-smooth | ✓ | ✓ | no need | no need |
| (Chen et al., 2023) | $1^{st}$ | $(L_0, L_1)$-smooth | ✗ | ✓ | no need | no need |
| (Ji et al., 2019b) | $0^{th}$ | $L$- smooth | ✓ | ✗ | no need | no need |
| (Huang et al., 2022) | $0^{th}$ | $L$- smooth | ✗ | ✓ | no need | need |
| (Xu et al., 2023) | $0^{th}$ | $L$- smooth | ✗ | ✓ | no need | need |
| (Liu et al., 2020) | $0^{th}$ | $L$- smooth | ✗ | ✓ | need | no need |
| **ZONSPIDER** (this paper) | $0^{th}$ | $(L_0, L_1)$-smooth | ✓ | ✓ | no need | no need |

Table 2: Query complexity comparison of the representative non-convex zeroth-order methods to find an $\epsilon$-stationary point of the black-box mini-optimization problems (1) and (2). **One estimator** denotes represent the number of function evaluations required to estimate a single gradient. **One iteration** denotes the number of gradient estimator required to update variable $x$. **Iteration complexity** denotes the total number of iterations required to find an $\epsilon$-stationary point.

| Problem | Method | one estimator | one iteration | Iteration complexity | Total function query Cost |
|---|---|---|---|---|---|
| **Finite-Sum** | (Huang et al., 2020) | $\mathcal{O}(d)$ | $\mathcal{O}(\sqrt{n})$ | $\mathcal{O}(\epsilon^{-2})$ | $\mathcal{O}(d\sqrt{n}\epsilon^{-2})$ |
| | **ZONSPIDER-coord**(this work) | $\mathcal{O}(d)$ | $\mathcal{O}(\sqrt{n})$ | $\mathcal{O}(\epsilon^{-2})$ | $\mathcal{O}(d\sqrt{n}\epsilon^{-2})$ |
| | **ZONSPIDER-rand**(this work) | $\mathcal{O}(d)$ | $\mathcal{O}(\sqrt{n})$ | $\mathcal{O}(\epsilon^{-2})$ | $\mathcal{O}(d\sqrt{n}\epsilon^{-2})$ |
| **General Expectation** | (Kornowski and Shamir, 2024) | $\mathcal{O}(1)$ | $\mathcal{O}(1)$ | $\tilde{\mathcal{O}}(d\epsilon^{-3})$ | $\tilde{\mathcal{O}}(d\epsilon^{-3})$ |
| | (Xu et al., 2023) | $\mathcal{O}(1)$ | $\mathcal{O}(\epsilon^{-1})$ | $\mathcal{O}(d\epsilon^{-2})$ | $\mathcal{O}(d\epsilon^{-3})$ |
| | **ZONSPIDER-coord**(this work) | $\mathcal{O}(d)$ | $\mathcal{O}(\epsilon^{-1})$ | $\mathcal{O}(\epsilon^{-2})$ | $\mathcal{O}(d\epsilon^{-3})$ |
| | **ZONSPIDER-rand**(this work) | $\mathcal{O}(d)$ | $\mathcal{O}(\epsilon^{-1})$ | $\mathcal{O}(\epsilon^{-3})$ | $\mathcal{O}(d\epsilon^{-3})$ |

## 2 PRELIMINARIES

Throughout the paper, $\|\cdot\|$ denotes the Euclidean norm for vectors, and operator norm for matrices, We use the symbol $\lfloor x \rfloor$ to denote the integer part of $x$.

**Assumption 1** (($L_0, L_1$)-smooth). *A differentiable function $f$ is said to be $(L_0, L_1)$-smooth if there exist constants $L_0 > 0, L_1 \geq 0$ such that if $\|x_1 - x_2\| \leq 1/L_1$, then*

$$\|\nabla f(x_1) - \nabla f(x_2)\| \leq (L_0 + L_1\|\nabla f(x_1)\|)\|x_1 - x_2\|.$$

*This also implies*

$$f(x_2) - f(x_1) - \langle \nabla f(x_1), x_2 - x_1 \rangle \leq \frac{(L_0 + L_1 \|\nabla f(x_1)\|)}{2} \|x_1 - x_2\|^2.$$

**Assumption 2** (Stochastic case). *In stochastic case, we need the following assumptions*

- *(i):In general expectation case, the stochastic oracle $f(x;\xi)$ is unbiased, i.e, : $\mathbb{E}[f(x;\xi)] = f(x)$, and $\mathbb{E}[\nabla f(x;\xi)] = \nabla f(x)$.*

- *(ii):We suppose variance of stochastic gradient is $(\sigma_0, \sigma_1)$-variance-bounded: $\mathbb{E}[\|\nabla f(x;\xi) - \nabla f(x)\|^2] \leq \sigma_0^2 + \sigma_1^2 \|\nabla f(x)\|^2$.*

- *(iii):For $\|x_1 - x_2\| \leq \frac{1}{2L_1}$, we suppose $(L_0, L_1)$-condition holds in stochastic case, in general expectation case, we suppose:*

$$\|\nabla f(x_1;\xi) - \nabla f(x_2;\xi)\| \leq (L_0 + L_1 \|\nabla f(x_1)\|) \|x_1 - x_2\|,$$

*in finite sum case:*

$$\|\nabla f_i(x_1) - \nabla f_i(x_2)\| \leq (L_0 + L_1 \|\nabla f(x_1)\|) \|x_1 - x_2\|.$$

**Remark 1.** *Instead of assuming traditional bounded variance assumption , we make a weaker assumption, called $(\sigma_0, \sigma_1)$- variance. Traditional bounded variance assumption is a special case that $\sigma_1 = 0$ (Xie et al., 2024; Chen et al., 2023), we emphasize that this assumption is only needed in general expectation case, we don't need this assumption in finite sum case.*

**Assumption 3.** *We suppose $f(x)$ has bounded minimum value $\Delta := f(x_0) - f(x^*) < \infty$.*

**Definition 1** ($\epsilon$-stationary point). *We say $x$ is an $\epsilon$-stationary point of $f(x)$ if $\|\nabla f(x)\| \leq \epsilon$ or $f(x) - f^* \leq \epsilon$.*

## 3 PROPOSED METHOD

In this section, we will introduce our method for solving both the finite-sum and expectation minimization problems. Firstly, we introduce the coord and rand zeroth-order gradient estimators and analyze the properties of these gradient operators under generalized-smooth conditions.

### 3.1 ZEROTH-ORDER GRADIENT ESTIMATOR

#### 3.1.1 RAND ESTIMATOR ANALYSIS UNDER GENERALIZED SMOOTHNESS

We first introduce smoothing function as follows:

$$f_\mu(x) := \mathbb{E}_{\{w \sim U_b\}}[f(x + \mu w)],$$

where $U_b$ is a uniform distribution over the unit Euclidean ball, following (Gao et al., 2017), its gradient can be expressed as $\nabla f_\mu(x) := \mathbb{E}_{\{v \sim U_{S_p}\}} \left[ \frac{n}{\mu} f(x + \mu v) v \right]$. Here $U_{S_p}$ is a uniform distribution over the unit Euclidean sphere, and $v \in \mathbb{R}^d$ is a random vector sampled from $U_{S_p}$.

We define zeroth-order rand estimator $\bar{\nabla} f(x)$ as follows, which is an unbiased estimator of $\nabla f_\mu(x)$:

$$\bar{\nabla} f(x) := \frac{d}{\mu}[f(x + \mu v) - f(x)]v, \qquad \text{(rand estimator)}$$

we also define the minibatch version of rand estimator using $S$ smoothing vector $v_j$:

$$\bar{\nabla}_S f(x) := \frac{1}{S} \sum_{j=1}^{S} \frac{d}{\mu}[f(x + \mu v_j) - f(x)]v_j, \qquad (4)$$

in stochastic case that we can't access to $f(x)$, we define the stochastic version of rand estimator in general expecation case and finite sum case:

$$\bar{\nabla}_S f(x; \xi) := \frac{1}{S} \sum_{j=1}^{S} \frac{d}{\mu}[f(x + \mu v_j; \xi) - f(x; \xi)]v_j, \bar{\nabla}_S f_i(x) := \frac{1}{S} \sum_{j=1}^{S} \frac{d}{\mu}[f_i(x + \mu v_j) - f_i(x)]v_j.$$

Rand estimator is an unbiased estimate of the gradient of the smoothing function(Gao et al., 2017), i.e, $\mathbb{E}[\bar{\nabla}_S f(x)] = \mathbb{E}[\bar{\nabla} f(x)] = \nabla f_\mu(x)$.

For rand estimator we have the following property: smoothing function := $f_\mu(x)$ is a good approximation of the original function $f(x)$, the error can be bounded by the following lemma.

**Lemma 1.** *Under assumption 1, we can bound the error between gradient of smoothing function $f_\mu(x)$ and the gradient of the original function $f$ as follows:*

$$\|\nabla f_\mu(x) - \nabla f(x)\|^2 \leq \frac{\mu^2 d^2 (L_0^2 + L_1^2 \|\nabla f(x)\|^2)}{2},$$

*The detailed proof is given in lemma D.1 of Appendix.*

Furthermore, the second-order moment of rand estimator can be bounded by the following lemma.

**Lemma 2.** *Under assumption 1, we can bound the second-order moment of the rand estimator $\bar{\nabla} f(x)$ as follows:*

$$\mathbb{E}_{\{v \sim U_{S_p}\}} \left[ \left\| \bar{\nabla} f(x) \right\|^2 \right] \leq 2d\|\nabla f(x)\|^2 + \frac{\mu^2 d^2 (L_0^2 + L_1^2 \|\nabla f(x)\|^2)}{2},$$

*The detailed proof is given in lemma D.2 of Appendix.*

The following lemma demonstrates the Lipschitz continuity(with a slight abuse of terminology) of the minibatch version of the rand estimator. To put it a bit less rigorously, we can say that the Lipschitz constant of the minibatch rand estimator scales as $\mathcal{O}\left(\sqrt{\frac{d}{S}}\right)$.

**Lemma 3.** *Under assumption 1, the Lipschitz property of the batch estimator $\bar{\nabla}_S f(x;\xi)$ is given as follows:*

$$\mathbb{E}[\|\bar{\nabla}_S f(x_1;\xi) - \bar{\nabla}_S f(x_2;\xi)\|^2] \leq 6\mu^2 d^2 L_0^2 + 9\mu^2 d^2 L_1^2 \|\nabla f(x_1)\|^2 + 3L_0^2(4+\frac{d}{S})\|x_1-x_2\|^2$$
$$+ (12L_0^2 + \frac{3}{2} + \frac{3dL_1^2}{S})\|\nabla f(x_1)\|^2 \|x_1-x_2\|^2.$$

*The detailed proof is given in lemma D.5 of Appendix.*

**Technical Novelty:** Compared with the original analysis of zeroth order method under standard smooth, we need to rebuild new approximate errors under $(L_0, L_1)$- smooth in Lemmas 1-3.

### 3.1.2 COORD ESTIMATOR ANALYSIS UNDER GENERALIZED SMOOTHNESS

**Definition 2** (Coord estimator)**.** *We define zeroth-order coord gradient estimator $\hat{\nabla}f(x)$ as follows:*

$$\hat{\nabla}f(x) := \sum_{\ell=1}^{d} \frac{1}{\mu}\left[f\left(x+\mu\mathbf{e}_\ell\right) - f\left(x\right)\right]\mathbf{e}_\ell, \qquad \text{(coord estimator)}$$

*where $\mathbf{e}_l$ is a standard basis vector with $1$ at its $l^{th}$ coordinate, and $0s$ elsewhere. In stochastic case that we can't access to $f(x)$, we define the stochastic version of coord estimator in general expecation case and finite sum case:*

$$\hat{\nabla}f(x;\xi) := \sum_{\ell=1}^{d} \frac{1}{\mu}\left[f\left(x+\mu\mathbf{e}_\ell;\xi\right) - f\left(x;\xi\right)\right]\mathbf{e}_\ell, \hat{\nabla}f_i(x) := \sum_{\ell=1}^{d} \frac{1}{\mu}\left[f_i\left(x+\mu\mathbf{e}_\ell\right) - f_i\left(x\right)\right]\mathbf{e}_\ell,$$
$$\tag{5}$$

*in finitesum case , we define it is easy to verify that $\mathbb{E}[\hat{\nabla}f(x;\xi)] = \hat{\nabla}f(x)$ and $\mathbb{E}[\hat{\nabla}f_i(x)] = \hat{\nabla}f(x)$.*

The following lemma provides an upper bound on the error of estimating $\nabla f(x)$ using $\hat{\nabla}f(x)$ under generalized-smooth conditions.

**Lemma 4.** *Under assumption 1, the following statement holds*

$$\left\|\hat{\nabla}f(x) - \nabla f(x)\right\| \leq \frac{L_0 + L_1\|\nabla f(x)\|}{2}\sqrt{d}\mu.$$

*The detailed proof is given in lemma C.1 of Appendix.*

The following lemma will show the generalized lipschitzness property of zeroth-order coord estimator.

**Lemma 5.** *Under assumptions 1 and 2, for $\|x_1 - x_2\| \leq \frac{2}{L_1}$ , we have*

$$\mathbb{E}\left[\left\|\hat{\nabla}f(x_1;\xi) - \hat{\nabla}f(x_2;\xi)\right\|^2\right] \leq 6(1 + L_1^2\mu^2 d^2)(L_0^2 + L_1^2\|\nabla f(x_1)\|^2)\|x_1-x_2\|^2 + 3L_0^2\mu^2 d^2$$
$$+ \frac{9}{2}L_1^2\mu^2 d^2\|\nabla f(x_1)\|^2$$

*The detailed proof is given in lemma C.3 of Appendix.*

### 3.2 ALGORITHM DESIGN

Equiped with zeroth-order gradient estimator, our method introduce SPIDER(Fang et al., 2018) and normalized step size into the zeroth-order gradient estimator and proposed zeroth-order normalized gradient method for solving both finite sum and general expectation optimizations. SPIDER is a variance reduction-typed method with optimal complexity guarantee, which uses large batch and small batch alternately to estimate stochastic gradients in a recursive way as follows

$$\mathbf{v}^k = \nabla f_B\left(\mathbf{x}^k\right) - \nabla f_B\left(\mathbf{x}^{k-1}\right) + \mathbf{v}^{k-1}, \quad \text{(SPIDER)}$$

Table 3: The different step-size design strategies, where $c$, $c_1$, and $c_2$ denote some constants.

| Method | stepsize | description |
|---|---|---|
| SPIDER(Fang et al., 2018) | $\eta_k = \min\{c_1, \frac{c_2 \epsilon}{\|v_k\|}\}$ | clipped stepsize |
| (Huang et al., 2022) | $\eta_k = \frac{c_1}{(c_2+k)^{1/3}}$ | diminishing stepsize |
| (Xu et al., 2023) | $\eta_k = o(\frac{1}{d})$ | constant stepsize |
| **ZONSPIDER** | $\eta_k = \frac{c\epsilon}{\|v_k\|}$ | normalized stepsize |

with clipped step size $\eta_k = \min\{c_1, \frac{c_2 \epsilon}{\|v_k\|}\}$ , where $c_1, c_2$ are some constants, and $\nabla f_B(x) = \frac{1}{|B|} \sum_{\xi \in B} \nabla f(x; \xi)$. Our method is a combination of SPIDER with normalized step size and zeroth-order gradient estimator, we call it ZONSPIDER, the main idea is to use coord or rand estimator to estimate the gradient of the original function, and use normalized stepsize to update the point $x$, the pseudo code is shown in algorithm 1. We compute the gradient estimator $v_k$ by sampling $B$ zeroth-order gradient estimator when $\mod(k, q) = 0$, and use small batch $b$ to compute the gradient estimator $v_k$ when $\mod(k, q) \neq 0$, and later update $x$ using normalized stepsize $x_{k+1} = x_k - \eta_k v_k$. The main difference between our algorithm and SPIDER is that SPIDER use clipped step-size, while we use a simpler normalized step-size $\eta_k = \frac{c_2 \epsilon}{\|v_k\|}$, as shown in Table 3.2. To address the additional challenges posed by the $(L_0, L_1)$-smooth condition, we adopt a different analytical approach from SPIDER, which is based on an inexact normalized descent lemma to obtain the decrease in function value (in expectation).

---

**Algorithm 1** ZO-normalized-SPIDER(ZONSPIDER)

---

**Initialization:** choose initialize point $x_0$, and $B, b, q$ as follows:

$$B = \begin{cases} \mathcal{O}(\epsilon^{-2} \max\{\sigma_0^2, \sigma_1^2\}) & \text{general expectation case} \\ n & \text{finite sum case} \end{cases}$$

$$b = \begin{cases} \epsilon^{-1} & \text{general expectation case} \\ \sqrt{n} & \text{finite sum case} \end{cases}$$

$q = b$

compute $v_0 = \frac{1}{B} \sum_{i=1}^{B} \hat{\nabla} f(x_0; \xi)$

**for** $k = 0, 1, \cdots, K-1$ **do**

  $\eta_k = \frac{c_2 \epsilon}{\|v_k\|}$

  $x_{k+1} = x_k - \eta_k v_k$

  **if** $\mod(k, q) = 0$ **then**

    Option I(coord): $v_{k+1} = \frac{1}{B} \sum_{i=1}^{B} \hat{\nabla} f(x_{k+1}; \xi_i)$(large batch) $\triangleright$ $\hat{\nabla} f(x_{k+1}; \xi)$(defied in (5))

    Option II(rand): $v_{k+1} = \frac{1}{B} \sum_{i=1}^{B} \bar{\nabla}_S f(x_{k+1}; \xi_i)$(large batch) $\triangleright$ $\bar{\nabla}_S f(x_{k+1}; \xi)$(defied in (4))

  **else**

    Option I(coord):$v_{k+1} = v_k + \frac{1}{b} \sum_{i=1}^{b} (\hat{\nabla} f(x_{k+1}; \xi_i) - \hat{\nabla} f(x_k; \xi_i))$(small batch)

    Option II(rand):$v_{k+1} = v_k + \frac{1}{b} \sum_{i=1}^{b} (\bar{\nabla}_S f(x_{k+1}; \xi_i) - \bar{\nabla}_S f(x_k; \xi_i))$(small batch)

  **end if**

**end for**

**return** (for theoretical) $x_\zeta$ chosen uniformly random from $\{x_k\}_{k=1}^{K}$.

**return** (for practical) $x_{K-1}$.

---

## 3.3 CONVERGENCE ANALYSIS

In this part, we give the convergence analysis of our method, we first introduce the inexact descent lemma, which is the key for our analysis. There are four theoretical results that need to be provided in this paper, we will give the analysis of coord estimator in finite sum (i.e. Theorem1) as an example.

**Lemma 6** (inexact descent lemma). *Under assumption 1 with $\eta_k = \frac{c_2\epsilon}{\|v_k\|}$, $c_2 \leq 1$, and $x_{k+1} - x_k = -\eta_k v_k$, we have:*

$$f(x_{k+1}) \leq f(x_k) - \left(c_2\epsilon - \frac{L_1 c_2^2 \epsilon^2}{2}\right) \|\nabla f(x_k)\| + 2c_2\epsilon \|v_k - \nabla f(x_k)\| + \frac{L_0 c_2^2 \epsilon^2}{2}. \quad (6)$$

*The detailed proof is given in lemma E.1 of Appendix.*

Next, to obtain the convergence rate, we need to estimate the term $\|v_k - \nabla f(x_k)\|$, we use the triangle inequality $\|v_k - \nabla f(x_k)\| \leq \left\|v_k - \hat{\nabla} f(x_k)\right\| + \left\|\hat{\nabla} f(x_k) - \nabla f(x_k)\right\|$, in lemma C.1 we have obtain the upperbound of $\left\|\hat{\nabla} f(x_k) - \nabla f(x_k)\right\|$, next we study the remaining term, $\|v_k - \nabla f(x_k)\|$, classical analysis of Sipder(Fang et al., 2018) often use the variance decomposition technique to obtain

$$\mathbb{E}\left[\left\|v_{k+1} - \hat{\nabla} f(x_{k+1})\right\|^2\right] \leq \mathbb{E}\left[\left\|v_k - \hat{\nabla} f(x_k)\right\|^2\right] + \frac{1}{b}\mathbb{E}\left[\left\|\hat{\nabla} f_i(x_{k+1}) - \hat{\nabla} f_i(x_k)\right\|^2\right],$$

thus summing up the above inequality from $k = \hat{k}$ to $k = q - 1$, we have

$$\mathbb{E}\left[\left\|v_k - \hat{\nabla} f(x_k)\right\|^2\right] \leq \frac{1}{b} \sum_{l=\hat{k}}^{\hat{k}+q-1} \mathbb{E}\left[\left\|\hat{\nabla} f_i(x_{k+1}) - \hat{\nabla} f_i(x_k)\right\|^2\right] + \underbrace{\mathbb{E}\left[\left\|v_{\hat{k}} - \hat{\nabla} f(x_{\hat{k}})\right\|^2\right]}_{=0(\text{finite sum case})},$$

in traditional $L$-smooth case, choosing parameters to let $\|x_{k+1} - x_k\| \leq \mathcal{O}(\epsilon)$, and let $q = b$, we can easily get upper bound

$$\mathbb{E}\left[\left\|v_k - \hat{\nabla} f(x_k)\right\|^2\right] \leq \mathcal{O}(\epsilon^2) + \underbrace{\mathbb{E}\left[\left\|v_{\hat{k}} - \hat{\nabla} f(x_{\hat{k}})\right\|^2\right]}_{=0(\text{finite sum case})} \leq \mathcal{O}(\epsilon^2), \quad (7)$$

but in $(L_0, L_1)$-smooth zeroth-order optimization, this equation contains addtional gradient norm terms as shown in the following lemma.

**Lemma 7** (Variance of finite sum case). *Under assumptions 1 and 2 , for Algorithm 1 with $\mu \leq \frac{1}{dL_1}$ we have*

$$\mathbb{E}\left[\left\|v_k - \hat{\nabla} f(x_k)\right\|\right] \leq 6L_0 c_2\epsilon + 2L_0 d\mu + \frac{1}{b}\sum_{l=\hat{k}}^{k}(6L_1 c_2\epsilon + 3L_1 d\mu)\|\nabla f(x_l)\|, \quad (8)$$

*The detailed proof is given in lemma E.2 of Appendix.*

From the above lemma, the variance term can't be bounded by a constans like equation(7), our stategy is taking (8) into lemma 6, and sum it from $k = \hat{k}$ to $k = \hat{k} + q$(one epoch) to obtain

$$\mathbb{E}[f(x_{q+\hat{k}}) - f(x_{\hat{k}})] \leq -\left(c_2\epsilon - \frac{L_1 c_2^2 \epsilon^2}{2} - L_1 c_2\epsilon\sqrt{d}\mu\right)\sum_{k=\hat{k}}^{q+\hat{k}-1}\|\nabla f(x_k)\|$$

$$+ \frac{2c_2\epsilon}{\sqrt{b}}\sum_{k=\hat{k}}^{\hat{k}+q-1}\sum_{l=\hat{k}}^{k}\left(4L_1 c_2\epsilon + 3L_1\sqrt{d}\mu\right)\|\nabla f(x_l)\| + \mathcal{O}(\epsilon^2),$$

a key observation is that in double sum term, every $\nabla f(x_l)$ appears at most $q$ times, this leads to $\sum_{k=\hat{k}}^{\hat{k}+q}\sum_{l=\hat{k}}^{k}(c_2\epsilon L_1 + 2d\mu L_1)\|\nabla f(x_l)\| \leq q\sum_{l=\hat{k}}^{q}\|\nabla f(x_l)\|$, and this terms be absorbed by the first term by the choice of parameters, thus we obtain the function value descent in $q$ itration:

$$\mathbb{E}[f(x_{q+\hat{k}}) - f(x_{\hat{k}})] \leq -\frac{qc_2\epsilon^2}{4},$$

this means we dereacse the function value by an average of $\frac{c_2\epsilon^2}{8}$ per iteration(in expectation), thus we need at most $K = \mathcal{O}(\Delta\epsilon^{-2})$ to find the stationary point, and the total number of oracle calls is

$$\#funtion = \mathcal{O}(d)K(b + \frac{B}{q}) = \mathcal{O}(d\epsilon^{-2}\sqrt{n}).$$

The following theorem is a formal statement of the above analysis.

**Theorem 1** (Finite sum case(coord estimator))**.** *For Algorithm 1 with coord estimator in finite sum case, under assumptions 1 and 2, let $c_2 \leq \min\{\frac{1}{72L_1}, \frac{1}{68L_0}\}$, choose $\eta_k = \frac{c_2\epsilon}{\|v_k\|}$, $\mu \leq \min\{\frac{\epsilon}{40\sqrt{d}L_0}, \frac{1}{56n^{\frac{1}{4}}L_1\sqrt{d}}\}$, $q = b = \sqrt{n}, B = n$, we have*

$$\mathbb{E}[f\left(x_{q+\hat{k}}\right) - f\left(x_{\hat{k}}\right)] \leq -\frac{qc_2\epsilon^2}{4}.$$

*We state that, in expectation, the function value of $f$ decreases by an average of $\frac{c_2\epsilon}{4}$ in, since $f(x)$ per iteration.Since $f$ can deacrease at most $\Delta$, we need at most*

$$K = \mathcal{O}(\Delta\epsilon^{-2}\max\{L_1, L_0\}),$$

*in expectation to find the stationary point, and the total numbers of the function query is*

$$\#function\ query = dT(b + \frac{B}{q}) = \mathcal{O}(d\epsilon^{-2}\sqrt{n}\max\{L_1, L_0\} + dn).$$

*The detailed proof is given in lemma E.1 of Appendix.*

We then give the results of other three cases as follows .

**Theorem 2** (Expectation case(coord estimator))**.** *For Algorithm 1 with coord estimator in expecation case, under assumptions 1 and 2, let $c_2 \leq \min\{\frac{1}{72L_1}, \frac{1}{68L_0}\}$, choose $\eta_k = \frac{c_2\epsilon}{\|v_k\|}$, $\mu \leq \min\{\frac{\epsilon}{56dL_0}, \frac{1}{56L_1\sqrt{d}\epsilon^{-0.5}}\}$, $B \geq \max\{\mathcal{O}(\epsilon^{-2}\sigma_1^2), \mathcal{O}(\epsilon^{-2}\sigma_0^2)\}$, $q = b = \epsilon^{-1}$, in expectation, we can find the stationary point in $K = \mathcal{O}(\Delta\epsilon^{-2}\max\{L_1, L_0\})$, and the total number of oracle calls $\#funtion = \mathcal{O}(d\epsilon^{-3}\max\{L_1, L_0\}\max\{\sigma_0^2, \sigma_1^2\} + dn\epsilon^{-2}\sigma_0^2)$.The detailed proof is given in lemma E.2 of Appendix.*

**Theorem 3** (Finite sum case(rand estimator))**.** *For Algorithm 1 with rand estimator in finite sum case, under assumptions 1 and 2, let $c_2 \leq \min\{\frac{1}{8(3L_1+2+4L_0)}, \frac{1}{36L_0}\}$, choose $\eta_k = \frac{c_2\epsilon}{\|v_k\|}$, $\mu \leq \min\{\frac{\epsilon}{40dL_0}, \frac{1}{20L_1d}\}$, $q = b = \sqrt{n}, B = n$, we need at most $K = \mathcal{O}(\Delta\epsilon^{-2}\max\{L_1, L_0\})$in expectation to find the stationary point, and the total number of e function query is $\#function\ query = dT(b + \frac{B}{q}) = \mathcal{O}(d\epsilon^{-2}\sqrt{n}\max\{L_1, L_0\} + dn)$. The detailed proof is given in lemma E.3 of Appendix.*

**Theorem 4** (Expectation case(rand estimator))**.** *For Algorithm 1 with rand estimator in expecation case, under assumptions 1 and 2, let $c_2 \leq \min\{\frac{1}{8(5L_1+2+4L_0)}, \frac{1}{36L_0}\}$, choose $\eta_k = \frac{c_2\epsilon}{\|v_k\|}$, $\mu \leq \min\{\frac{\epsilon}{40dL_0}, \frac{1}{20L_1d}\}$, $q = b = \epsilon^{-1}$, $B \geq \max\{\mathcal{O}(\epsilon^{-2}(3 + \sigma_1)^2), \mathcal{O}(\epsilon^{-2}\sigma_0^2)\}$, we need at most $K = \mathcal{O}(\Delta\epsilon^{-2}\max\{L_1, L_0\})$ in expectation to find the stationary point, and the total number of e function query is $\#funtion = \mathcal{O}(d)K(b + \frac{B}{q}) = \mathcal{O}(d\epsilon^{-3}\max\{\sigma_1^2, \sigma_0^2\}\max\{L_1, L_0\} + \epsilon^2\max\{\sigma_1^2, \sigma_0^2\})$.The detailed proof is given in lemma E.4 of Appendix.*

## 4 EXPERIMENTS

We conduct two experiments to verify the effectiveness of our method: the first experiment focuses on Phase Retrieval, while the second examines Distributionally Robust Optimization (DRO), as detailed in (Chen et al., 2023). In Phase Retrieval, we first analyze the effects of different parameters of the rand and coord estimators, presented in Figures 1(a) and 1(b). Subsequently, we compare the effectiveness of poposed ZONSPIDER method against other first-order algorithms in both Phase Retrieval and DRO, shown in Figures 1(c) and 1(d). Notably, we use sample complexity to measure the cost; for zeroth-order algorithms, sample complexity refers to the number of zeroth-order gradient estimators utilized.

## 4.1 APPLICATION TO NONCONVEX PHASE RETRIEVAL

Phase retrieval is a well-known nonconvex problem in machine learning and signal processing(Miao et al., 1999). Let $x \in \mathbb{R}^d$ represent the true underlying object, and assume we collect $m$ intensity measurements, given by $y_r = |\mathbf{a}_r^\top x|^2$ for $r = 1, 2, \ldots, m$, where $\mathbf{a}_r \in \mathbb{R}^d$. The challenge in phase retrieval lies in recovering the signal by solving the associated nonconvex optimization problem:

$$\min_{z \in \mathbb{R}^d} f(z) := \frac{1}{2m} \sum_{r=1}^m \left( y_r - |\mathbf{a}_r^\top z|^2 \right)^2 . \tag{10}$$

The above nonconvex objective function is a high-order polynomial in the high-dimensional space. Therefore, it does not belong to the $L$-smooth function class $\mathcal{L}$.

We evaluate the performance of our algorithms by applying them to the non-convex phase retrieval problem described in (10).We adopt the same setup as in (Chen et al., 2023), and we refer readers to appendix A for more details about hyper-parameters.

First, to provide insight into the parameters used in the zeroth-order estimator, we compare the effects of different values of $S$ in the rand estimator, as shown in Figure 1(a). We observe that choosing $S = d$ negatively impacts the performance of the rand estimator, while $S = 10d$ and $S = 50d$ yield similar results. Thus, we believe the ideal range is $d < S \leq 10d$. Next, we examine the effects of different smoothing parameters on both the rand and coord estimators, with results presented in Figure 1(b). The smoothing parameter proves to be quite robust, as selecting $\mu \leq 10^{-3}$ suffices to achieve good performance for both estimators. Finally, we compare the performance of different algorithms in Phase Retrieval, with results displayed in Figure 1(c). We note that (i) ZONSPIDER-coord and SPIDER demonstrate the best performance, and (ii) the coord estimator exhibits more stable performance compared to the rand estimator.

## 4.2 APPLICATION TO DISTRIBUTIONAL ROBUST OPTIMIZATION

Distributional Robust Optimization (DRO) is a widely used framework for training robust models, Under mild conditions, it aims to solve the following problem:

$$\min_{x \in \mathcal{X}, \eta \in \mathbb{R}} L(x, \eta) := \lambda \mathbb{E} \xi \sim P \psi^* \left( \frac{\ell \xi(x) - \eta}{\lambda} \right) + \eta \tag{9}$$

where $\psi^*$ is the convex conjugate of $\psi$, and we refer readerser to appendix A for more details about hyperparameters. We solve the non-convex DRO problem (9) using life expectancy data, which includes 2,413 samples of life expectancy and influencing factors. After preprocessing (e.g., filling missing values, standardizing variables), we use 2,000 samples for training, with features $x_i \in \mathbb{R}^{34}$ and target $y_i \in \mathbb{R}$, we set $\lambda = 0.01$ and use the $\chi^2$ divergence for $\psi^*(t) = \frac{1}{4}(t+2)^2 - 1$. The regularized mean square loss function is:$\ell_\xi(w) = \frac{1}{2}(y_\xi - x_\xi^\top w)^2 + 0.1 \sum_{j=1}^{34} \ln \left( 1 + |w^{(j)}| \right)$, initialize $\eta_0 = 0.1$ and $w_0 \in \mathbb{R}^{34}$ randomly using a Gaussian distribution.

We compare the performance of several algorithms. The results in Figure 1(d) lead to similar conclusions as those from the Phase Retrieval experiment, namely: (i) ZONSPIDER-coord and its first-order variant perform the best, and (ii) the coordinate estimator outperforms the random estimator.

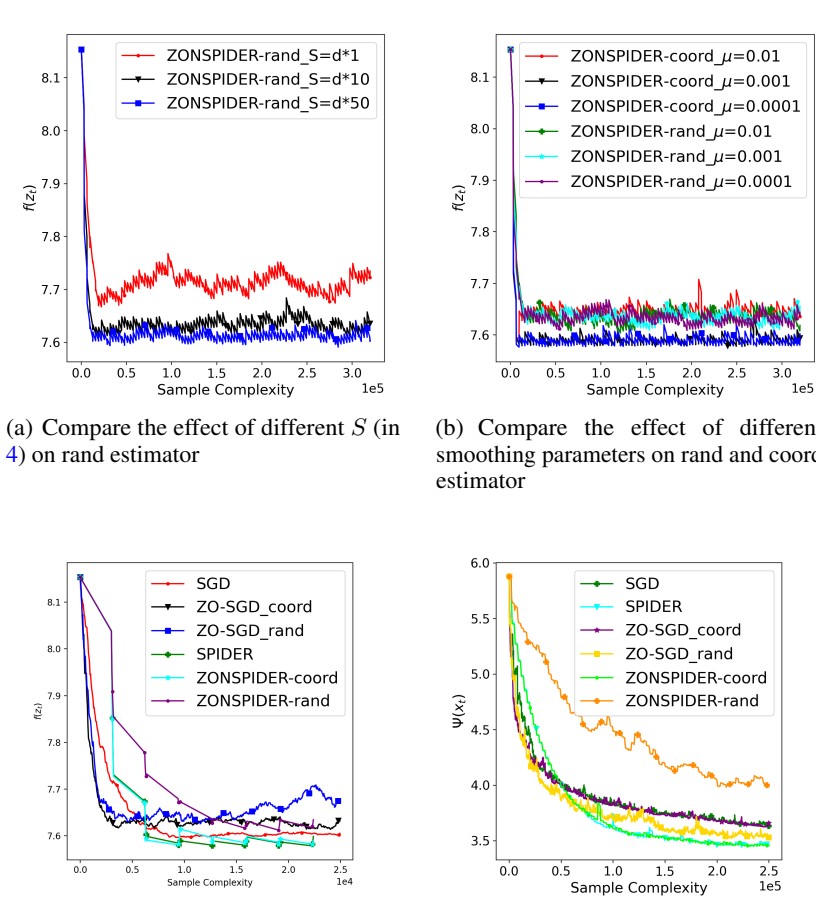

(a) Compare the effect of different $S$ (in 4) on rand estimator

(b) Compare the effect of different smoothing parameters on rand and coord estimator

(c) Compare different algorithms on Phase Retrieval

(d) Compare different algorithms on DRO

Figure 1: Experiments results

## 5 CONCLUSION

In this paper, we address the question of whether zeroth-order methods can be safely applied to problems that exhibit $(L_0, L_1)$-smoothness. We propose a variance-reduced zeroth-order method called ZONSPIDER, a variant of SPIDER (Fang et al., 2018), which utilizes normalized stepsizes and zeroth-order gradient estimators. We provide an analysis of both coordand rand estimators under the finite sum and general expectation cases, showing that the total number of function value queries required to obtain an $\epsilon$-stationary point is upper bounded by $\mathcal{O}(d\epsilon^{-2})$ and $\mathcal{O}(d\epsilon^{-3})$, respectively. To the best of our knowledge, this is the first application of zeroth-order methods to $(L_0, L_1)$-smooth problems. A further direction for research is to explore whether zeroth-order methods can be safely applied to additional problems under the $(L_0, L_1)$-smooth condition, such as $PL$-conditions, strongly convex conditions, and general convex conditions.

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

CONTENTS

Table 4: Meaning of symbols

| Symbol | Description |
|---|---|
| $\hat{\nabla}f(x)$ | coord estimator |
| $\hat{\nabla}f(x;\xi)$ | coord estimator using sample function $f(x;\xi)$ |
| $\nabla f_\mu(x)$ | gradient of smoothing function |
| $\bar{\nabla}f_\mu(x)$ | rand gradient estimator |
| $\bar{\nabla}_S f_\mu(x)$ | average rand estimator of $S$ times |
| $\bar{\nabla}_S f_\mu(x)$ | average rand estimator of $S$ times using sample function $f(x;\xi)$ |
| $d$ | dimension of problem |
| $L_0, L_1$ | generalized Lipschitz constant |
| $\mu$ | smoothing parameter |
| $v$ | smoothing vector |
| $\mathbf{n}$ | total number of data points(finite sum case) |
| $\sigma_0, \sigma_1$ | $\mathbb{E}[\|\nabla f(x;\xi) - \nabla f(x)\|^2] \leq \sigma_0^2 + \sigma_1^2\|\nabla f(x)\|^2$ (general expectation case) |
| $\mathbf{e}_\ell$ | the $l$-th basis vector of $R^n$ |
| $b$ | batch size when $mod(k,q) \neq 0$ |
| $B$ | batch size when $mod(k,q) = 0$ |
| $\hat{k}$ | $\hat{k} = \lfloor k/q \rfloor q$ |
| $q$ | a parameter that control the frequence of use large batchsize $B$ |

# A HYPERPARAMETERS DETAILS

## A.1 PHASE RETRIEVAL

We choose the problem dimension to be $d = 100$ and the sample size to be $m = 3000$. The measurement vectors $a_r \in \mathbb{R}^d$ and the true parameter $z \in \mathbb{R}^d$ are generated element-wise from a Gaussian distribution $\mathcal{N}(0, 0.5)$. For the initialization, $z_0 \in \mathbb{R}^d$ is drawn element-wise from $\mathcal{N}(5, 0.5)$. The measurements are then constructed as $y_i = |a_r^T z|^2 + n_i$ for $i = 1, \ldots, m$, where the noise term $n_i$ is sampled from $\mathcal{N}(0, 4^2)$, representing additive Gaussian noise.

We set the parameters for the basic SGD with $\gamma = 2 \times 10^{-4}$, normalized SGD with $\gamma = 2 \times 10^{-3}$, and normalized SGD with momentum, where $\gamma = 3 \times 10^{-3}$ and the momentum coefficient is $10^{-4}$. For clipped SGD, we set $\gamma = 0.3$ and use a normalization term $\max(\|\nabla f(z_t)\|, 10^3)$. For SPIDER, the learning rate is $\gamma = 0.01$, the epoch size is $q = 5$, and the batch sizes are $B = 3000$ and $B' = 50$. For the proposed ZONSPIDER method, we choose $\mu = 10^{-4}$ for both the coord and rand estimators, $S = 10d$ for the rand estimator, and the remaining parameters are the same as those in SPIDER.

## A.2 DRO

We set the standard SGD with $\gamma = 2 \times 10^{-4}$, normalized SGD with $\gamma = 8 \times 10^{-3}$, and normalized SGD with momentum, where $\gamma = 8 \times 10^{-3}$ and the momentum coefficient is $10^{-4}$. For clipped SGD, $\gamma = 0.05$ with a normalization term $\max(|\nabla L(x_t, \eta_t)|, 100)$, and for SPIDER, the learning rate is $\gamma = 4 \times 10^{-3}$, the epoch size is $q = 20$, and the batch sizes are $B = 2000$ and $B' = 50$. The initialization is obtained by running normalized GD with $\gamma = 0.2$ for 30 iterations starting from $w_0, \eta_0$. For the proposed ZONSPIDER method, we choose $\mu = 10^{-4}$ for the coord estimator, $\mu = 0.005$, and $S = 13d$ for the randestimator. The other parameters are the same as those in SPIDER.

# B   AUXILARY LEMMA

**Lemma B.1** (Jensen's inequality). *For convex function $f(x)$ we have*

$$f(\mathbb{E}[x]) \leq \mathbb{E}[f(x)], \tag{10}$$

*two extended versions of Jensen's inequality are*

$$\|\mathbb{E}[x]\| \leq \mathbb{E}[\|x\|], \text{ for } x \in \mathbb{R}^d$$

$$\left\|\sum_{i=1}^{k} a_i\right\|^2 \leq k \sum_{i=1}^{k} \|a_i\|^2, \text{ for } a_i \in \mathbb{R}^d.$$

**Lemma B.2** (Young's inequality). *For any vectors $a, b, \in \mathbb{R}^d$, and $\zeta \geq 0$, the following inequality holds:*

$$\|a\|^2 \leq (1+\zeta)\|a-b\|^2 + \left(1+\zeta^{-1}\right)\|b\|^2,$$

*an extended version of Young's inequality is*

$$\langle a, b \rangle \leq \frac{\|a\|^2}{2\zeta} + \frac{\zeta\|b\|^2}{2}.$$

**Lemma B.3** (Variance decomposition). *For random vector $x \in \mathbb{R}^d$ and any $y \in \mathbb{R}^d$, the variance of $x$ can be decomposed as*

$$\mathbb{E}\left[\|x - \mathbb{E}[x]\|^2\right] = \mathbb{E}\left[\|x - y\|^2\right] - \mathbb{E}\left[\|\mathbb{E}[x] - y\|^2\right],$$

*which implies*

$$\mathbb{E}\left[\|x - \mathbb{E}[x]\|^2\right] \leq \mathbb{E}\left[\|x\|^2\right].$$

**Lemma B.4.** *For random variable $X, Y$, if $X, Y$ are independent, and $\mathbb{E}[X]$ or $\mathbb{E}[Y] = 0$, we have*

$$\mathbb{E}[\|X - Y\|^2] = \mathbb{E}[\|X\|^2] + \mathbb{E}[\|Y\|^2], \tag{11}$$

*Proof.*

$$\mathbb{E}[\|X - Y\|^2] = \mathbb{E}[\|X\|^2 + \|Y\|^2 + 2\mathbb{E}\langle X, Y\rangle] = \mathbb{E}[\|X\|^2] + \mathbb{E}[\|Y\|^2].$$

$\square$

**Lemma B.5.** *For i.i.d. $x_1, x_2, x_3 \cdots x_n$, if they satifisies $\mathbb{E}[x_i] = x$, we have*

$$\mathbb{E}\left[\left\|\frac{1}{b}\sum_{i=1}^{b} x_i - x\right\|^2\right] \leq \frac{\mathbb{E}[\|x_i\|^2]}{b}. \tag{12}$$

*Proof.*

$$\mathbb{E}\left[\left\|\frac{1}{b}\sum_{i=1}^{b} x_i - x\right\|^2\right]$$

$$= \frac{1}{b^2}\mathbb{E}\left[\left\|\sum_{i=1}^{b}(x_i - x)\right\|^2\right]$$

$$= \frac{1}{b^2}\sum_{i=1}^{b}\mathbb{E}[\|x_i - x\|^2]$$

$$= \frac{1}{b}\mathbb{E}[\|x_i - x\|^2] \leq \frac{\mathbb{E}[\|x_i\|^2]}{b}.$$

where the second inequality holds due to $\|a+b\|^2 = \|a\|^2 + \|b\|^2 + 2\langle a, b\rangle$, and $\mathbb{E}[\langle x_i - x, x_j - x\rangle] = 0 (j \neq i)$ for iid random variable $x_i$. $\square$

**Lemma B.6** ((Gao et al., 2017)). *Let $\alpha(n)$ be the volume of the unit ball in $\mathbb{R}^n$, and $\beta(n)$ be the surface area of the unit sphere in $\mathbb{R}^n$. We also denote $B$, and $S_p$, to be the unit ball and unit sphere respectively. Let $I$ be the identity matrix in $\mathbb{R}^{n \times n}$, then*

$$\int_{S_p} vv^\top dv = \frac{\beta(n)}{n} I.$$

## C  COORD ESTIMATOR ANALYSIS UNDER GENERALIZED SMOOTHNESS

The following lemma upper bound the error between gradient estimator $\hat{\nabla} f(x)$ and $\nabla f(x)$.

**Lemma C.1** (Restatement of lemma 4). *Under assumption 1, we have*

$$\left\| \hat{\nabla} f(x) - \nabla f(x) \right\| \leq \frac{L_0 + L_1 \|\nabla f(x)\|}{2} \sqrt{d}\mu.$$

*Proof.*

$$\left\| \hat{\nabla} f(x) - \nabla f(x) \right\|^2 = \left\| \sum_{\ell=1}^d \left( \frac{\partial f_\mu(x)}{\partial x_\ell} - \frac{\partial f(x)}{\partial x_\ell} \right) \mathbf{e}_\ell \right\|^2$$

$$= \sum_{\ell=1}^d \left\| \frac{\partial f_\mu(x)}{\partial x_\ell} - \frac{\partial f(x)}{\partial x_\ell} \right\|^2 \leq \frac{(L_0 + L_1 \|\nabla f(x)\|)^2}{4} d\mu^2,$$

where the second inequality holds due to $\left\| \frac{\partial f_\mu(x)}{\partial x_\ell} - \frac{\partial f(x)}{\partial x_\ell} \right\| = \left\| \frac{f(x+\mu \mathbf{e}_\ell) - f(x) - \langle \nabla f(x), \mu e_\ell \rangle}{\mu} \right\| \leq \frac{\mu(L_0 + L_1 \|\nabla f(x)\|)}{2}$.

Note that we also assume $\nabla f(x; \xi)$ is $(L_0, L_1)$-smooth, so we also have

$$\left\| \hat{\nabla} f(x; \xi) - \nabla f(x; \xi) \right\|^2 \leq \frac{(L_0 + L_1 \|\nabla f(x)\|)^2}{4} d\mu^2.$$

$\square$

The following lemma will upper bound the error between stochastic zeroth-order estimator and true estimator.

**Lemma C.2.** *Under assumptions 1 and 2 , we have*

$$\mathbb{E}\left[ \left\| \hat{\nabla} f(x; \xi) - \hat{\nabla} f(x) \right\|^2 \right] \leq 3(L_0^2 + L_1^2 \|\nabla f(x)\|^2)\mu^2 d + 3\sigma_0^2 + 3\sigma_1^2 \|\nabla f(x)\|^2. \tag{13}$$

*Proof.*

$$\mathbb{E}\left[ \left\| \hat{\nabla} f(x; \xi) - \hat{\nabla} f(x) \right\|^2 \right]$$

$$\leq 3 \left\| \hat{\nabla} f(x; \xi) - \nabla f(x; \xi) \right\|^2 + 3 \left\| \nabla f(x) - \hat{\nabla} f(x) \right\|^2 + 3\mathbb{E}\left[ \|\nabla f(x; \xi) - \nabla f(x)\|^2 \right]$$

$$\overset{a}{\leq} 3(L_0^2 + L_1^2 \|\nabla f(x)\|^2) d\mu^2 + 3\sigma_0^2 + 3\sigma_1^2 \|\nabla f(x)\|^2.$$

where (a) holds due to the conclusion of lemma C.1 , and assumption 2. $\square$

The following lemma will show the Lipschitzness property of stochastic zeroth-order estimator.

**Lemma C.3** (Restatement of lemma 5). *Under assumptions 1 and 2, let $\mu \leq \frac{1}{\sqrt{d}L_1}$, for $\|x_1 - x_2\| \leq \frac{2}{L_1}$, we have*

$$\mathbb{E}\left[ \left\| \hat{\nabla} f(x_1; \xi) - \hat{\nabla} f(x_2; \xi) \right\|^2 \right] \leq 15(L_0^2 + L_1^2 \|\nabla f(x_1)\|^2) \|x_1 - x_2\|^2 + 3d\mu^2 L_0^2 + \frac{9}{2} d\mu^2 L_1^2 \|\nabla f(x_1)\|^2,$$

*and for finite sum case, we have the same conclusion:*

$$\mathbb{E}\left[ \left\| \hat{\nabla} f_i(x_1) - \hat{\nabla} f_i(x_2) \right\|^2 \right] \leq 15(L_0^2 + L_1^2 \|\nabla f(x_1)\|^2) \|x_1 - x_2\|^2 + 3d\mu^2 L_0^2 + \frac{9}{2} d\mu^2 L_1^2 \|\nabla f(x_1)\|^2.$$

*Proof.*

$$\mathbb{E}[\left\|\hat{\nabla}f(x_1;\xi) - \hat{\nabla}f(x_2;\xi)\right\|^2]$$

$$\leq 3\mathbb{E}\left[\|\nabla f(x_1;\xi) - \hat{\nabla}f(x_1;\xi)\|^2\right] + 3\left\|\nabla f(x_2;\xi) - \hat{\nabla}f(x_2;\xi)\right\|^2 + 3\left\|\nabla f(x_2;\xi) - \hat{\nabla}f(x_2;\xi)\right\|^2$$

$$\overset{a}{\leq} 12(L_0^2 + L_1^2 \|\nabla f(x_1)\|^2)\|x_1 - x_2\|^2 + 3d\mu^2(L_0^2 + \frac{1}{2}L_1^2\left(\|\nabla f(x_1)\|^2 + \|\nabla f(x_2)\|^2\right))$$

$$\overset{b}{\leq} 12(L_0^2 + L_1^2 \|\nabla f(x_1)\|^2)\|x_1 - x_2\|^2 + 3d\mu^2 L_0^2 + \frac{3}{2}d\mu^2 L_1^2(3\|\nabla f(x_1)\|^2 + 2\|\nabla f(x_1) - \nabla f(x_2)\|^2)$$

$$\overset{c}{\leq} 15(L_0^2 + L_1^2 \|\nabla f(x_1)\|^2)\|x_1 - x_2\|^2 + 3d\mu^2 L_0^2 + \frac{9}{2}d\mu^2 L_1^2 \|\nabla f(x_1)\|^2.$$

where step(a) holds due to the conclusion of lemma C.1, and assumption 2.

step (b) holds due to $\|\nabla f(x_2)\|^2 \leq 2(\|\nabla f(x_1)\|^2 + \|\nabla f(x_1) - \nabla f(x_2)\|^2)$.

step(c) holds due to $((L_0, L_1)$-smooth)-smooth and we let $\mu \leq \frac{1}{\sqrt{d}L_1}$.

note that in finite sum case, we can use the same proof to obtain the same conclusion. $\square$

## D    RAND ESTIMATOR ANALYSIS UNDER GENERALIZED SMOOTHNESS

**Lemma D.1** (Restatement of lemma 1). *Under assumption 1, we can bound the error between gradient of smoothing function $f_\mu$ and the gradient of the original function $f$ as follows:*

$$\|\nabla f_\mu(x) - \nabla f(x)\|^2 \leq \frac{\mu^2 d^2(L_0^2 + L_1^2 \|\nabla f(x)\|^2)}{2}. \tag{14}$$

*Proof.*

$$\|\nabla f_\mu(x) - \nabla f(x)\|$$

$$= \left\|\frac{1}{\beta(d)}\left[\frac{d}{\mu}\int_{S_p} f(x + \mu v)v\,dv\right] - \nabla f(x)\right\|$$

$$\overset{lemma B.6}{=} \left\|\frac{1}{\beta(d)}\left[\frac{d}{\mu}\int_{S_p} f(x + \mu v)v\,dv - \int_{S_p}\frac{d}{\mu}f(x)v\,dv - \int_{S_p}\frac{d}{\mu}\langle\nabla f(x), \mu v\rangle v\,dv\right]\right\|$$

$$\leq \frac{d}{\beta(d)\mu}\int_{S_p}|f(x + \mu v) - f(x) - \langle\nabla f(x), \mu v\rangle|\|v\|\,dv$$

$$\leq \frac{d}{\beta(d)\mu}\frac{(L_0 + L_1\|\nabla f(x)\|)\mu^2}{2}\int_{S_p}\|v\|^3\,dv = \frac{\mu d(L_0 + L_1\|\nabla f(x)\|)}{2}.$$

$\square$

**Lemma D.2** (Restatement of lemma 2). *Under assumption 1, we can bound the second-order moment of the rand estimator $\bar{\nabla}f(x)$ as follows:*

$$\mathbb{E}_{\{v\sim U_{S_p}\}}\left[\|\bar{\nabla}f(x)\|^2\right] \leq 2d\|\nabla f(x)\|^2 + \frac{\mu^2 d^2}{2}(L_0 + L_1\|\nabla f(x)\|)^2. \tag{15}$$

*Proof.*

$$\mathbb{E}_{\{v \sim U_{S_p}\}}\left[\left\|\bar{\nabla}f(x)\right\|^2\right] = \frac{1}{\beta(d)}\int_{S_p}\frac{d^2}{\mu^2}|f(x+\mu v)-f(x)|^2\|v\|^2 dv$$

$$= \frac{d^2}{\beta(d)\mu^2}\int_{S_p}[f(x+\mu v)-f(x)-\langle\nabla f(x),\mu v\rangle+\langle\nabla f(x),\mu v\rangle]^2 dv$$

$$\leq \frac{d^2}{\beta(d)\mu^2}\int_{S_p}\left[2(f(x+\mu v)-f(x)-\langle\nabla f(x),\mu v\rangle)^2+2(\langle\nabla f(x),\mu v\rangle)^2\right]dv$$

$$\leq \frac{d^2}{\beta(d)\mu^2}\left[\int_{S_p}2\left(\frac{(L_0+L_1\|\nabla f(x)\|)\mu^2}{2}\|v\|^2\right)^2 dv+2\mu^2\int_{S_p}\nabla f(x)^\top vv^\top\nabla f(x)dv\right]$$

$$\overset{lemma B.6}{=} \frac{d^2}{\beta(d)\mu^2}\left[\frac{(L_0+L_1\|\nabla f(x)\|)^2\mu^4}{2}\beta(d)+2\mu^2\frac{\beta(d)}{d}\|\nabla f(x)\|^2\right]$$

$$= 2d\|\nabla f(x)\|^2+\frac{\mu^2 d^2}{2}(L_0+L_1\|\nabla f(x)\|)^2.$$

$\square$

**Lemma D.3** (variance decompsition). *for constant c we have*

$$\mathbb{E}\left[\|x-\mathbb{E}[x]\|^2\right] = \mathbb{E}\left[\|x-c\|^2\right]-\|\mathbb{E}[x]-c\|^2. \tag{16}$$

**Lemma D.4.** *Under assumption 1, we can bound the error between batch estimator and the gradient of the original function as follows:*

$$\mathbb{E}\left[\|\bar{\nabla}_S f(x)-\nabla f(x)\|^2\right] \leq (\mu^2 d^2 L_1^2+\frac{2d}{S})\|\nabla f(x)\|^2+\mu^2 d^2 L_0^2. \tag{17}$$

*Proof.* from lemma D.1 and D.3 and we have

$$\|\nabla f_\mu(x)-\nabla f(x)\|^2 = \left\|\mathbb{E}[\bar{\nabla}f(x)]-\nabla f(x)\right\|^2$$

$$= \mathbb{E}\left[\|\bar{\nabla}f(x)-\nabla f(x)\|^2\right]-\mathbb{E}\|\mathbb{E}[\bar{\nabla}f(x)]-\bar{\nabla}f(x)\|^2$$

$$\leq (\frac{\mu d(L_0+L_1\|\nabla f(x)\|)}{2})^2,$$

then, combine lemma D.2, we obtain:

$$\mathbb{E}\left[\|\bar{\nabla}_S f(x)-\nabla f(x)\|^2\right] \leq (\frac{\mu d(L_0+L_1\|\nabla f(x)\|)}{2})^2+\mathbb{E}\|\mathbb{E}[\bar{\nabla}_S f(x)]-\nabla f_\mu(x)\|^2$$

$$\leq (\frac{\mu d(L_0+L_1\|\nabla f(x)\|)}{2})^2+\frac{1}{S}\mathbb{E}\left[\|\bar{\nabla}f(x)\|^2\right]$$

$$\leq \frac{\mu^2 d^2(L_0^2+L_1^2\|\nabla f(x)\|^2)}{2}+\frac{2d}{S}\|\nabla f(x)\|^2+\frac{\mu^2 d^2}{2S}(L_0+L_1\|\nabla f(x)\|)^2$$

$$\overset{S\geq 2}{\leq} (\mu^2 d^2 L_1^2+\frac{2d}{S})\|\nabla f(x)\|^2+\mu^2 d^2 L_0^2,$$

where the second inequality holds due to lemma B.5, i.e.

$$\mathbb{E}\|\mathbb{E}[\bar{\nabla}_S f(x)]-\nabla f_\mu(x)\|^2 \leq \frac{1}{S}\mathbb{E}\left[\|\bar{\nabla}f(x)-\nabla f_\mu(x)\|^2\right] \leq \frac{1}{S}\mathbb{E}\left[\|\bar{\nabla}f(x)\|^2\right],$$

where the third inequality holds due to lemma D.2. $\square$

**Lemma D.5** (Restatement of lemma 3). *Under assumption 1, we can give the Lipschitzness(with a little abuse of terminology) of the batch estimator $\bar{\nabla}_S f(x)$ as follows:*

$$\mathbb{E}\left[\left\|\bar{\nabla}f(x_1;\xi)-\bar{\nabla}f(x_2;\xi)\right\|^2\right] \leq 3d^2 L_0^2\mu^2+3dL_0^2\|x_1-x_2\|^2+\frac{9d^2 L_1^2\mu^2}{2}\|\nabla f(x_1)\|^2$$

$$+(\frac{3\mu^2 d^2 L_1^2}{2}+3dL_1^2)\|\nabla f(x_1)\|^2\|x_1-x_2\|^2,$$

*and for minibatch smoothing estimator, we also have a similar conclusion, but the difference is that the some coefficient were divided by $S$:*

$$\mathbb{E}\left[\left\|\bar{\nabla}_S f(x_1;\xi) - \bar{\nabla}_S f(x_2;\xi)\right\|^2\right] \leq 6\mu^2 d^2 L_0^2 + 9\mu^2 d^2 L_1^2 \left\|\nabla f(x_1)\right\|^2 + 3L_0^2(4 + \frac{d}{S})\left\|x_1 - x_2\right\|^2$$
$$+ (12L_0^2 + \frac{3}{2} + \frac{3dL_1^2}{S})\left\|\nabla f(x_1)\right\|^2 \left\|x_1 - x_2\right\|^2.$$

*Proof.*

$$\mathbb{E}\left[\left\|\bar{\nabla} f(x_1;\xi) - \bar{\nabla} f(x_2;\xi)\right\|^2\right]$$
$$= d^2\mathbb{E}[\|\frac{v}{\mu}[f(x_1 + \mu v;\xi) - f(x_1;\xi) - \langle\nabla f(x_1;\xi), v\rangle]v - \frac{v}{\mu}[f(x_2 + \mu v) - f(x_2) - \langle\nabla f(x_2;\xi), v\rangle]$$
$$+ (\langle\nabla f(x_1;\xi), v\rangle v - \langle\nabla f(x_2;\xi), v\rangle v)\|^2]$$
$$\overset{a}{\leq} d^2(3L_0^2\mu^2 + \frac{3L_1^2\mu^2(\|\nabla f(x_1)\|^2 + \|\nabla f(x_2)\|^2)}{2} + \mathbb{E}\left[3\|\langle\nabla f(x_1;\xi), v\rangle v - \langle\nabla f(x_2;\xi), v\rangle v\|^2\right]$$
$$\overset{b}{\leq} d^2(3L_0^2\mu^2 + \frac{3L_1^2\mu^2(3 + 4L_1^2\|x_1 - x_2\|^2)\|\nabla f(x_1)\|^2}{2} + \mathbb{E}\left[3\|\langle\nabla f(x_1;\xi), v\rangle v - \langle\nabla f(x_2;\xi), v\rangle v\|^2\right]$$
$$\overset{c}{\leq} d^2(3L_0^2\mu^2 + \frac{3L_1^2\mu^2(3 + 4L_1^2\|x_1 - x_2\|^2)\|\nabla f(x_1)\|^2}{2}) + \mathbb{E}\left[3d\|\nabla f(x_2;\xi) - \nabla f(x_1;\xi)\|^2\right]$$
$$\leq 3d^2 L_0^2\mu^2 + 3dL_0^2\|x_1 - x_2\|^2 + \frac{9d^2 L_1^2\mu^2}{2}\|\nabla f(x_1)\|^2 + (\frac{3d^2 L_1^2\mu^2}{2} + 3dL_1^2)\|\nabla f(x_1)\|^2\|x_1 - x_2\|^2,$$
$$\tag{18}$$

where step(a) holds due to $\|a + b + c\|^2 \leq 3(\|a\|^2 + \|b\|^2 + \|c\|^2)$ and $f(x_1;\xi) \leq f(x_1;\xi) + \langle\nabla f(x_1;\xi), x_2 - x_1\rangle + \frac{1}{2}(L_0 + L_1\|\nabla f(x_1;\xi)\|)\|x_2 - x_1\|^2$.

step(b) holds due to $\|\nabla f(x_2)\|^2 \leq 2(\|\nabla f(x_1)\|^2 + \|\nabla f(x_2) - \nabla f(x_1)\|^2) \leq 2(\|\nabla f(x_1)\|^2 + 2(L_0^2 + 2L_1^2\|\nabla f(x_1)\|^2)\|x_1 - x_2\|^2)$

and in step(c), denote $y = \nabla f(x_1;\xi) - \nabla f(x_2;\xi)$, then $\|\langle\nabla f(x_1;\xi), v\rangle v - \langle\nabla f(x_2;\xi), v\rangle v\|^2 = y^T v v^T y \leq \|y\|^2 \mathbb{E}[\|v v^T\|]$, since $v$ is a vector randomly sampled from the unit sphere, from ((Ji et al., 2019b) lemma5) we know $\mathbb{E}[\|v v^T\|] = \frac{1}{d}I_d$, thus $\mathbb{E}[\|\langle\nabla f(x_1;\xi), v\rangle v - \langle\nabla f(x_2;\xi), v\rangle v\|^2] \leq \frac{1}{d}\|\nabla f(x_2;\xi) - \nabla f(x_1;\xi)\|^2$.

Next, we proceed to prove the second part.

$$\mathbb{E}\left[\left\|\bar{\nabla}_S f(x_1;\xi) - \bar{\nabla}_S f(x_2;\xi)\right\|^2\right] = \mathbb{E}\left[\left\|\frac{1}{S}\sum_{j=1}^{S}\bar{\nabla}_j f(x_1;\xi) - \bar{\nabla}_j f(x_2;\xi)\right\|^2\right]$$

$$=\mathbb{E}\left[\left\|\frac{1}{S}\sum_{j=1}^{S}\bar{\nabla}_j f(x_1;\xi) - \bar{\nabla}_j f(x_2;\xi) \pm \nabla f_\mu(x_1;\xi) \pm \nabla f_\mu(x_2;\xi)\right\|^2\right]$$

$$=\mathbb{E}\left[\left\|\frac{1}{S}\sum_{j=1}^{S}\bar{\nabla}_j f(x_1;\xi) - \bar{\nabla}_j f(x_2;\xi) + \nabla f_\mu(x_1;\xi) - \nabla f_\mu(x_1;\xi)\right\|^2\right] + \mathbb{E}\left[\left\|\nabla f_\mu(x_1;\xi) - \nabla f_\mu(x_2;\xi)\right\|^2\right]$$

$$\overset{lemma B.5}{\leq}\frac{1}{S}\mathbb{E}\left[\left\|\bar{\nabla}_j f(x_1;\xi) - \bar{\nabla}_j f(x_2;\xi)\right\|^2\right] + \mathbb{E}\left[\left\|\nabla f_\mu(x_1;\xi) - \nabla f_\mu(x_2;\xi)\right\|^2\right]$$

$$\leq\frac{1}{S}\mathbb{E}\left[\left\|\bar{\nabla}_j f(x_1;\xi) - \bar{\nabla}_j f(x_2;\xi)\right\|^2\right] + 3\mathbb{E}[\|\nabla f(x_1;\xi) - \nabla f_\mu(x_1;\xi)\|^2]$$
$$+ 3\mathbb{E}[\|\nabla f(x_2;\xi) - \nabla f_\mu(x_2;\xi)\|^2] + 3\mathbb{E}[\|\nabla f(x_1;\xi) - \nabla f(x_2;\xi)\|^2]$$

$$\leq\frac{1}{S}\mathbb{E}\left[\left\|\bar{\nabla}_j f(x_1;\xi) - \bar{\nabla}_j f(x_2;\xi)\right\|^2\right]$$
$$+ 3\frac{\mu^2 d^2(L_0^2 + L_1^2\|\nabla f(x_1)\|^2)}{2} + 3\frac{\mu^2 d^2(L_0^2 + L_1^2\|\nabla f(x_2)\|^2)}{2} + 6(L_0^2 + L_1^2\|\nabla f(x_1)\|^2)\|x_1 - x_2\|^2$$

$$\leq\frac{1}{S}\mathbb{E}\left[\left\|\bar{\nabla}_j f(x_1;\xi) - \bar{\nabla}_j f(x_2;\xi)\right\|^2\right] + 3\frac{\mu^2 d^2(L_0^2 + L_1^2\|\nabla f(x_1)\|^2)}{2}$$
$$+ 3\frac{\mu^2 d^2(L_0^2 + 2L_1^2\left(\|\nabla f(x_1))\|^2 + \|\nabla f(x_1) - \nabla f(x_2)\|^2\right))}{2} + 6(L_0^2 + L_1^2\|\nabla f(x_1)\|^2)\|x_1 - x_2\|^2$$

$$\overset{a}{\leq}\frac{1}{S}\mathbb{E}\left[\left\|\bar{\nabla}_j f(x_1;\xi) - \bar{\nabla}_j f(x_2;\xi)\right\|^2\right] + 3\mu^2 d^2 L_0^2 + \frac{9\mu d^2 L_1^2}{2}\|\nabla f(x_1)\|^2$$
$$+ 6L_0^2(\mu^2 d^2 L_1^2 + 1)\|x_1 - x_2\|^2 + 6L_0^2(\mu^2 d^2 L_1^2 + 1)\|\nabla f(x_1)\|^2\|x_1 - x_2\|^2$$

$$\overset{b}{\leq}6\mu^2 d^2 L_0^2 + 9d^2 L_1^2\mu^2\|\nabla f(x_1)\|^2 + 3L_0^2(4 + \frac{d}{S})\|x_1 - x_2\|^2 + (12L_0^2 + \frac{3}{2} + \frac{3dL_1^2}{S})\|\nabla f(x_1)\|^2\|x_1 - x_2\|^2,$$

where (a) holds due to $(L_0, L_1)$ smoothness,

step(b) holds due to (18) and we let $S \geq 1$, $\mu \leq \frac{1}{dL_1}$. $\qquad\square$

In the following lemma, we analyze the variance introduced by sampling data points.

**Lemma D.6.** *Under assumptions 1 and 2, we have*

$$\mathbb{E}\left\|\bar{\nabla}_S f(x;\xi) - \bar{\nabla}_S f(x)\right\|^2 \leq 6(\mu^2 d^2 L_1^2 + \frac{2d}{S} + \sigma_1^2)\|\nabla f(x)\|^2 + 6\mu^2 d^2 L_0^2 + 6\sigma_0^2.$$

*Proof.*

$$\mathbb{E}\left\|\bar{\nabla}_S f(x;\xi) - \bar{\nabla}_S f(x)\right\|^2$$

$$\leq 3\mathbb{E}\left[\left\|\bar{\nabla}_S f(x;\xi) - \nabla f(x;\xi)\right\|^2\right] + 3\mathbb{E}\left[\left\|\bar{\nabla}_S f(x) - \nabla f(x)\right\|^2\right] + 3\mathbb{E}\left[\left\|\nabla f(x) - \nabla f(x;\xi)\right\|^2\right]$$

$$\leq 6(\mu^2 d^2 L_1^2 + \frac{2d}{S})\|\nabla f(x)\|^2 + 6\mu^2 d^2 L_0^2 + 6(\sigma_0^2 + \sigma_1^2\|\nabla f(x)\|^2)$$

$$= 6(\mu^2 d^2 L_1^2 + \frac{2d}{S} + \sigma_1^2)\|\nabla f(x)\|^2 + 6\mu^2 d^2 L_0^2 + 6\sigma_0^2.$$

$\qquad\square$

# E  CONVERGENCE ANALYSIS

**Lemma E.1** (Restatement of lemma 6). *Let $\eta_k = \frac{c_2\epsilon}{\|v_k\|}$, $c_2 \leq 1$, under assumption 1, for $x_{k+1} - x_k = -\eta_k v_k$, we have:*

$$f(x_{k+1}) \leq f(x_k) - \left(c_2\epsilon - \frac{L_1 c_2^2 \epsilon^2}{2}\right) \|\nabla f(x_k)\| + 2c_2\epsilon \|v_k - \nabla f(x_k)\| + \frac{L_0 c_2^2 \epsilon^2}{2}.$$

*Proof.* since , $\eta_k = \frac{c_2\epsilon}{\|v_k\|}$, from $(L_0, L_1)$ smooth, we have:

$$
\begin{aligned}
f(x_{k+1}) &\leq f(x_k) + \langle \nabla f(x_k), x_{k+1} - x_k \rangle + \frac{L_0 + L_1 \|\nabla f(x_k)\|}{2} \|x_{k+1} - x_k\|^2 \\
&= f(x_k) - c_2\epsilon \left\langle \nabla f(x_k), \frac{v_k}{\|v_k\|} \right\rangle + c_2^2 \epsilon^2 \frac{L_0 + L_1 \|\nabla f(x_k)\|}{2} \\
&= f(x_k) - c_2\epsilon \|v_k\| + c_2\epsilon \left\langle v_k - \nabla f(x_k), \frac{v_k}{\|v_k\|} \right\rangle + c_2^2 \epsilon^2 \frac{L_0 + L_1 \|\nabla f(x_k)\|}{2} \\
&\leq f(x_k) - c_2\epsilon \|v_k\| + c_2\epsilon \|v_k - \nabla f(x_k)\| + c_2^2 \epsilon^2 \frac{L_0 + L_1 \|\nabla f(x_k)\|}{2} \\
&\overset{a}{\leq} f(x_k) - c_2\epsilon \|\nabla f(x_k)\| + 2c_2\epsilon \|v_k - \nabla f(x_k)\| + c_2^2 \epsilon^2 \frac{L_0 + L_1 \|\nabla f(x_k)\|}{2} \\
&\leq f(x_k) - \left(c_2\epsilon - \frac{L_1 c_2^2 \epsilon^2}{2}\right) \|\nabla f(x_k)\| + 2c_2\epsilon \|v_k - \nabla f(x_k)\| + \frac{c_2^2 \epsilon^2 L_0}{2}.
\end{aligned}
$$

(a) is due to $\|v_k\| \geq \|\nabla f(x_k)\| - \|v_k - \nabla f(x_k)\|$. $\qquad\square$

## E.1  CONVERGENCE ANALYSIS OF COORD ESTIMATOR

In the following lemma, we disscuss the behavior of variance term $\left\|v_k - \hat{\nabla} f(x_k)\right\|^2$ in finite sum case.

**Lemma E.2** (Restatement of lemma 7). *Under assumptions 1 and 2 , for Algorithm 1, choose $\mu \leq \frac{1}{\sqrt{d}L_1}$, $B = n$, denote $\hat{k} = \lfloor k/q \rfloor q$ , we have*

$$\mathbb{E}\left[\left\|v_k - \hat{\nabla} f(x_k)\right\|\right] \leq 4L_0 c_2 \epsilon + 2\sqrt{d}\mu L_0 + \frac{1}{\sqrt{b}} \sum_{l=\hat{k}}^{k-1} \left(4L_1 c_2 \epsilon + 3\sqrt{d}\mu L_1\right) \|\nabla f(x_l)\|.$$

*Proof.* note that $\mathbb{E}[\hat{\nabla} f(x_k; \xi)] = \hat{\nabla} f(x_k)$

$$
\mathbb{E}\left[\left\|v_{k+1} - \hat{\nabla} f(x_{k+1})\right\|^2\right]
$$

$$
\leq \mathbb{E}\left[\left\|v_k + \frac{1}{b}\sum_{i=1}^{b}(\hat{\nabla} f_i(x_{k+1}) - \hat{\nabla} f_i(x_k)) \pm \hat{\nabla} f(x_k) - \hat{\nabla} f(x_{k+1})\right\|^2\right]
$$

$$
\overset{a}{\leq} \mathbb{E}\left[\left\|v_k - \hat{\nabla} f(x_k)\right\|^2\right] + \mathbb{E}\left[\left\|\frac{1}{b}\sum_{i=1}^{b}(\hat{\nabla} f_i(x_{k+1}) - \hat{\nabla} f_i(x_k)) + \hat{\nabla} f(x_k) - \hat{\nabla} f(x_{k+1})\right\|^2\right]
$$

$$
\overset{b}{\leq} \mathbb{E}\left[\left\|v_k - \hat{\nabla} f(x_k)\right\|^2\right] + \frac{1}{b}\mathbb{E}\left[\left\|\hat{\nabla} f_i(x_{k+1}) - \hat{\nabla} f_i(x_k)\right\|^2\right]
$$

$$
\overset{c}{\leq} \mathbb{E}\left[\left\|v_k - \hat{\nabla} f(x_k)\right\|^2\right] + \frac{1}{b}\left(15(L_0^2 + L_1^2 \|\nabla f(x_k)\|^2)\|x_k - x_{k+1}\|^2 + 3d\mu^2 L_0^2 + \frac{9}{2}d\mu^2 L_1^2 \|\nabla f(x_k)\|^2\right),
$$

where step (a) holds due to $\mathbb{E}[\frac{1}{b}\sum_{i=1}^{b}(\hat{\nabla} f_i(x_{k+1}) - \hat{\nabla} f_i(x_k)) + \hat{\nabla} f(x_k) - \hat{\nabla} f(x_{k+1})] = 0$ and lemma B.4.

step(b) holds due to lemma B.5.($\mathbb{E}\left[\left\|\frac{1}{b}\sum_{i=1}^{b}x_i - x\right\|^2\right] \leq \frac{\mathbb{E}[\|x_i\|^2]}{b}$).

step(c) holds due to lemma C.3 to upper bound $\mathbb{E}\left[\left\|\hat{\nabla}f_i(x_{k+1}) - \hat{\nabla}f_i(x_k)\right\|^2\right]$.

Now, note that (i)$\|x_{k+1} - x_k\| \leq c_2\epsilon$ due to the choice of $\eta_k$, (ii) for Algorithm 1, in finite sum case, when $k = \hat{k}$, we use all $n$ samples to compute the gradient, which means $v_{\hat{k}} = \hat{\nabla}f(x_{\hat{k}})$, so $\mathbb{E}\left[\left\|v_{\hat{k}} - \hat{\nabla}f(x_{\hat{k}})\right\|^2\right] = \mathbb{E}\left[\left\|\hat{\nabla}f(x_{\hat{k}}) - \hat{\nabla}f(x_{\hat{k}})\right\|^2\right] = 0$, we obtain

$$\mathbb{E}\left[\left\|v_k - \hat{\nabla}f(x_k)\right\|^2\right]$$

$$\leq \frac{1}{b}\sum_{l=\hat{k}}^{k-1}\left(15(L_0^2 + L_1^2\|\nabla f(x_l)\|^2)\|x_{l+1} - x_l\|^2 + 3d\mu^2 L_0^2 + \frac{9}{2}d\mu^2 L_1^2\|\nabla f(x_l)\|^2\right) + \underbrace{\mathbb{E}\left[\left\|v_{\hat{k}} - \hat{\nabla}f(x_{\hat{k}})\right\|^2\right]}_{=0(\text{finite sum case})}$$

$$\leq \frac{q}{b}\left(15L_0^2 c_2^2\epsilon^2 + 3L_0^2 d\mu^2\right) + \frac{1}{b}\sum_{l=\hat{k}}^{k-1}\left(15L_1^2 c_2^2\epsilon^2 + \frac{9}{2}L_1^2 d\mu^2\right)\|\nabla f(x_l)\|^2, \tag{19}$$

let $q = b$, use the fact that $\sqrt{\sum\|x_i\|^2} \leq \sum\|x_i\|$ when every $x_i \geq 0$, and $\mathbb{E}[\|x\|] \leq \sqrt{\mathbb{E}[\|x\|^2]}$, we have

$$\mathbb{E}\left[\left\|v_k - \hat{\nabla}f(x_k)\right\|\right] \leq \sqrt{15}L_0 c_2\epsilon + \sqrt{3d}\mu L_0 + \frac{1}{\sqrt{b}}\sum_{l=\hat{k}}^{k-1}\left(\sqrt{15}L_1 c_2\epsilon + \frac{3}{\sqrt{2}}\sqrt{d}\mu L_1\right)\|\nabla f(x_l)\|.$$

To improve readability, we perform some scaling on square root terms and let $\mu \leq \frac{1}{\sqrt{d}L_1}$, this leads to:

$$\mathbb{E}\left[\left\|v_k - \hat{\nabla}f(x_k)\right\|\right] \leq 4L_0 c_2\epsilon + 2\sqrt{d}\mu L_0 + \frac{1}{\sqrt{b}}\sum_{l=\hat{k}}^{k-1}\left(4L_1 c_2\epsilon + 3\sqrt{d}\mu L_1\right)\|\nabla f(x_l)\|.$$

$\square$

Furthermore, We disscuss the behavior of variance term $\|v_k - \hat{\nabla}f(x_k)\|^2$ in Expectation case.

**Lemma E.3.** *Under assumptions 1 and 2 , for Algorithm 1, choose $q = b$, $\mu \leq \frac{1}{dL_1}$, denote $\hat{k} = \lfloor k/q \rfloor q$, we have*

$$\mathbb{E}\left[\left\|v_k - \hat{\nabla}f(x_k)\right\|\right] \leq 4L_0 c_2\epsilon + 3\sqrt{d}\mu L_0 + \frac{2\sigma_0}{\sqrt{B}} + \sum_{l=\hat{k}}^{k-1}\left(\frac{1}{\sqrt{b}}\left(4L_1 c_2\epsilon + 3\sqrt{d}\mu L_1\right) + \frac{2L_1\mu\sqrt{d} + 2\sigma_1}{\sqrt{B}}\right)\|\nabla f(x_l)\|.$$

*Proof.* Similar to the previous theorem of finite sum case, but the difference is that in Expectation case, $\mathbb{E}[\left\|\frac{1}{B}(\hat{\nabla}f(x_{\hat{k}};\xi)) - \hat{\nabla}f(x_{\hat{k}})\right\|^2] \neq 0$

$$\mathbb{E}\left[\left\|v_k - \hat{\nabla}f(x_k)\right\|^2\right]$$

$$\leq \frac{1}{b}\sum_{l=\hat{k}}^{k-1}\left(15(L_0^2 + L_1^2\|\nabla f(x_l)\|^2)\|x_{l+1} - x_l\|^2 + 3d\mu^2 L_0^2 + \frac{9}{2}d\mu^2 L_1^2\|\nabla f(x_l)\|^2\right) + \mathbb{E}\left[\left\|v_{\hat{k}} - \hat{\nabla}f(x_{\hat{k}})\right\|^2\right]$$

$$\leq \frac{q}{b}\left(15L_0^2 c_2^2\epsilon^2 + 3L_0^2 d\mu^2\right) + \frac{1}{b}\sum_{l=\hat{k}}^{k-1}\left(15L_1^2 c_2^2\epsilon^2 + \frac{9}{2}L_1^2 d\mu^2\right)\|\nabla f(x_l)\|^2 \tag{20}$$

$$+ \frac{1}{B}\mathbb{E}\left[\left\|\hat{\nabla}f(x_{\hat{k}};\xi) - \hat{\nabla}f(x_{\hat{k}})\right\|^2\right],$$

where the second inequality holds due to that fact that when $k = \hat{k}$, $v_{\hat{k}} = \frac{1}{B} \sum_{i=1}^{B} \hat{\nabla} f(x_k; \xi_i)$, and from lemma C.2 we have $\mathbb{E}\left[\left\|\hat{\nabla} f(x_{\hat{k}}; \xi) - \hat{\nabla} f(x_{\hat{k}})\right\|^2\right] \leq 3(L_0^2 + L_1^2 \left\|\nabla f(x)\right\|^2)\mu^2 d + 3\sigma_0^2 + 3\sigma_1^2 \left\|\nabla f(x)\right\|^2$. let $q = b$, $\mu \leq \frac{1}{dL_1}$, we have

$$\mathbb{E}\left[\left\|v_k - \hat{\nabla} f(x_k)\right\|^2\right]$$

$$\leq \left(15L_0^2 c_2^2 \epsilon^2 + 3L_0^2 d\mu^2\right) + \frac{1}{b} \sum_{l=\hat{k}}^{k-1} \left(15L_1^2 c_2^2 \epsilon^2 + \frac{9}{2}L_1^2 d\mu^2\right) \left\|\nabla f(x_l)\right\|^2$$

$$+ \frac{1}{B}\left(3(L_0^2 + L_1^2 \left\|\nabla f(x_{\hat{k}})\right\|^2)\mu^2 d + 3\sigma_0^2 + 3\sigma_1^2 \left\|\nabla f(x_{\hat{k}})\right\|^2\right),$$

now, use the fact that $\sqrt{\sum \left\|x_i\right\|^2} \leq \sum \left\|x_i\right\|$ when every $x_i \geq 0$, and $\mathbb{E}[\left\|x\right\|] \leq \sqrt{\mathbb{E}[\left\|x\right\|^2]}$, and $B \gg 1$ we obtain

$$\mathbb{E}\left[\left\|v_k - \hat{\nabla} f(x_k)\right\|\right] \leq 4L_0 c_2 \epsilon + 3\sqrt{d}\mu L_0 + \frac{2\sigma_0}{\sqrt{B}} + \sum_{l=\hat{k}}^{k-1}\left(\frac{1}{\sqrt{b}}\left(4L_1 c_2 \epsilon + 3\sqrt{d}\mu L_1\right) + \frac{2L_1\mu\sqrt{d} + 2\sigma_1}{\sqrt{B}}\right)\left\|\nabla f(x_l)\right\|.$$

$\square$

**Theorem E.1** (Restatement of Theorem 1 (coord estimator in finite sum case)). *For Algorithm 1, under assumptions 1 and 2, let $c_2 \leq \min\{\frac{1}{72L_1}, \frac{1}{68L_0}\}$, choose $\eta_k = \frac{c_2\epsilon}{\|v_k\|}$, $\mu \leq \min\{\frac{\epsilon}{40\sqrt{d}L_0}, \frac{1}{56n^{\frac{1}{4}}L_1\sqrt{d}}\}$, $q = b = \sqrt{n}$, $B = n$, we have*

$$\mathbb{E}[f\left(x_{q+\hat{k}}\right) - f\left(x_{\hat{k}}\right)] \leq -\frac{qc_2\epsilon^2}{4}.$$

*We state that, in expectation, the function value of $f$ decreases by an average of $\frac{c_2\epsilon}{4}$ in, since $f(x)$ per iteration. Since $f$ can deacrease at most $\Delta$, we need at most*

$$K = \mathcal{O}(\Delta\epsilon^{-2} \max\{L_1, L_0\}),$$

*in expectation to find the stationary point, and the total number of e function query is*

$$\#function \; query = dT(b + \frac{B}{q}) = \mathcal{O}(d\epsilon^{-2}\sqrt{n} \max\{L_1, L_0\} + dn).$$

*Proof.* From the conclusion of lemma E.1, and upper bound of $\left\|\nabla f(x_k) - \hat{\nabla} f(x_k)\right\|^2$ (lemma C.1), we have

$$f\left(x_{k+1}\right) \leq f\left(x_k\right) - \left(c_2\epsilon - \frac{c_2^2\epsilon^2}{2}L_1\right)\left\|\nabla f\left(x_k\right)\right\| + 2c_2\epsilon\left\|v_k - \nabla f\left(x_k\right)\right\| + \frac{c_2^2\epsilon^2 L_0}{2}$$

$$\leq f\left(x_k\right) - \left(c_2\epsilon - \frac{c_2^2\epsilon^2}{2}L_1\right)\left\|\nabla f\left(x_k\right)\right\|$$

$$+ 2c_2\epsilon\left(\left\|\nabla f(x_k) - \hat{\nabla} f(x_k)\right\| + \left\|v_k - \hat{\nabla} f(x_k)\right\|\right) + \frac{c_2^2\epsilon^2 L_0}{2}$$

$$\leq f\left(x_k\right) - \left(c_2\epsilon - \frac{c_2^2\epsilon^2}{2}L_1\right)\left\|\nabla f\left(x_k\right)\right\|$$

$$+ 2c_2\epsilon\left(\left(\frac{L_0 + L_1\left\|\nabla f(x)\right\|}{2}\sqrt{d}\mu\right) + \left\|v_k - \hat{\nabla} f(x_k)\right\|\right) + \frac{c_2^2\epsilon^2 L_0}{2},$$

then from upper bound of $\mathbb{E}\left[\left\|v_k - \hat{\nabla} f(x_k)\right\|^2\right]$ (lemma E.2), we have $\mathbb{E}\left[\left\|v_k - \hat{\nabla} f(x_k)\right\|\right] \leq$

$4L_0 c_2 \epsilon + 2\sqrt{d}\mu L_0 + \frac{1}{\sqrt{b}} \sum_{l=\hat{k}}^{k-1} \left(4L_1 c_2 \epsilon + 3\sqrt{d}\mu L_1\right) \|\nabla f(x_l)\|.$

$$\mathbb{E}[f(x_{k+1}) - f(x_k)] \leq -\left(c_2\epsilon - \frac{L_1 c_2^2 \epsilon^2}{2} - L_1 c_2 \epsilon \sqrt{d}\mu\right)\|\nabla f(x_k)\|$$

$$+ \frac{2c_2\epsilon}{b}\sum_{l=\hat{k}}^{k}\left(4L_1 c_2\epsilon + 3L_1\sqrt{d}\mu\right)\|\nabla f(x_l)\| + \frac{17L_0 c_2^2 \epsilon^2}{2} + 5L_0 c_2\epsilon\sqrt{d}\mu,$$

sum it from $k = \hat{k}$ to $q$, we obtain

$$\mathbb{E}[f\left(x_{q+\hat{k}}\right) - f(x_{\hat{k}})]$$

$$\leq -\left(c_2\epsilon - \frac{L_1 c_2^2 \epsilon^2}{2} - L_1 c_2\epsilon\sqrt{d}\mu\right)\sum_{k=\hat{k}}^{q+\hat{k}-1}\|\nabla f(x_k)\| + \frac{2c_2\epsilon}{\sqrt{b}}\sum_{k=\hat{k}}^{\hat{k}+q-1}\sum_{l=\hat{k}}^{k}\left(4L_1 c_2\epsilon + 3L_1\sqrt{d}\mu\right)\|\nabla f(x_l)\|$$

$$+ q\left(\frac{17L_0 c_2^2 \epsilon^2}{2} + 5L_0 c_2\epsilon\sqrt{d}\mu\right)$$

$$\overset{a}{\leq} -c_2\epsilon\left(1 - L_1 c_2\epsilon(\frac{1}{2} + 8\sqrt{q}) - (1 + 6\sqrt{q})L_1\sqrt{d}\mu\right)\sum_{k=\hat{k}}^{q+\hat{k}-1}\|\nabla f(x_l)\| + qc_2\epsilon\left(\frac{17L_0 c_2\epsilon}{2} + 5L_0\sqrt{d}\mu\right)$$

$$\overset{b}{\leq} -\frac{c_2\epsilon}{2}\sum_{k=\hat{k}}^{q+\hat{k}-1}\|\nabla f(x_l)\| + \frac{qc_2\epsilon^2}{4}$$

$$\overset{c}{\leq} -\frac{qc_2\epsilon^2}{4}, \tag{21}$$

step(a) holds due to the observation that $\sum_{k=\hat{k}}^{\hat{k}+q}\sum_{l=\hat{k}}^{k}(c_2\epsilon L_1 + 2d\mu L_1)\|\nabla f(x_l)\| \leq q\sum_{l=\hat{k}}^{q}\|\nabla f(x_l)\|$

step(b) holds because we let choose $q = b = \sqrt{n}$ and we suppose $\epsilon \leq \frac{1}{n^{0.25}}$, we choose $c_2 \leq \min\{\frac{1}{72L_1}, \frac{1}{68L_0}\}$ to let $L_1 c_2\epsilon(\frac{1}{2} + 8\sqrt{q}) \leq 9L_1 c_2\epsilon \leq \frac{1}{8}$ and $\frac{17c_2\epsilon L_0}{2} \leq \frac{\epsilon}{8}$, $\mu \leq \min\{\frac{\epsilon}{40\sqrt{d}L_0}, \frac{1}{56n^{\frac{1}{4}}L_1\sqrt{d}}\}$ to let $(1 + 6\sqrt{q})L_1\sqrt{d}\mu \leq 7n^{\frac{1}{4}}L_1\sqrt{d}\mu \leq \frac{1}{8}$ and $5d\mu L_0 \leq \frac{\epsilon}{8}$

step(c) holds due to $\|\nabla f(x_k)\| \geq \epsilon$ otherwise we have find the stationary point.

Now, from(21), we know that in the sense of expecation, $f(x)$ descrease at least $\frac{qc_2\epsilon^2}{4}$ in $q$ steps, that is, $\frac{c_2\epsilon^2}{4}$ every step in average, since $f(x)$ can deacrease at most $\Delta$, in the sense of expectation we need at most

$$K = \mathcal{O}(\Delta\epsilon^{-2}\max\{L_1, L_0\})$$

to find the stationary point, and the total number of function query is

$$\#funtion = dT(b + \frac{B}{q}) = \mathcal{O}(d\epsilon^{-2}\sqrt{n}\max\{L_1, L_0\} + dn).$$

$\square$

**Theorem E.2** (Restatement of Theorem 2 (coord estimator in expecation case)). *For Algorithm 1, under assumptions 1 and 2, let $c_2 \leq \min\{\frac{1}{72L_1}, \frac{1}{68L_0}\}$, choose $\eta_k = \frac{c_2\epsilon}{\|v_k\|}$, $\mu \leq \min\{\frac{\epsilon}{56dL_0}, \frac{1}{56L_1\sqrt{d}\epsilon^{-0.5}}\}$, $B \geq \max\{\mathcal{O}(\epsilon^{-2}\sigma_1^2), \mathcal{O}(\epsilon^{-2}\sigma_0^2)\}$, $q = b = \epsilon^{-1}$, we have*

$$\mathbb{E}[f\left(x_{q+\hat{k}}\right) - f(x_{\hat{k}})] \leq -\frac{qc_2\epsilon^2}{4},$$

*in expectation, we can find the stationary point in $K = \mathcal{O}(\Delta\epsilon^{-2}\max\{L_1, L_0\}$, and the total number of oracle calls*

$$\#funtion = dT(b + \frac{B}{q}) = \mathcal{O}(d\epsilon^{-3}\max\{L_1, L_0\}\max\{\sigma_0^2, \sigma_1^2\} + dn\epsilon^{-2}\sigma_0^2).$$

*Proof.* From the conclusion of lemma E.1, and upper bound of $\left\|\nabla f(x_k) - \hat{\nabla} f(x_k)\right\|^2$ (lemma C.1), we have

$$
\begin{aligned}
f\left(x_{k+1}\right) \leq & f\left(x_k\right) - \left(c_2\epsilon - \frac{c_2^2\epsilon^2}{2}L_1\right)\left\|\nabla f\left(x_k\right)\right\| + 2c_2\epsilon\left\|v_k - \nabla f\left(x_k\right)\right\| + \frac{c_2^2\epsilon^2 L_0}{2} \\
\leq & f\left(x_k\right) - \left(c_2\epsilon - \frac{c_2^2\epsilon^2}{2}L_1\right)\left\|\nabla f\left(x_k\right)\right\| \\
& + 2c_2\epsilon\left(\left\|\nabla f(x_k) - \hat{\nabla} f(x_k)\right\| + \left\|v_k - \hat{\nabla} f(x_k)\right\|\right) + \frac{c_2^2\epsilon^2 L_0}{2} \\
\leq & f\left(x_k\right) - \left(c_2\epsilon - \frac{c_2^2\epsilon^2}{2}L_1\right)\left\|\nabla f\left(x_k\right)\right\| \\
& + 2c_2\epsilon\left(\left(\frac{L_0 + L_1\left\|\nabla f(x)\right\|}{2}d\mu\right) + \left\|v_k - \hat{\nabla} f(x_k)\right\|\right) + \frac{c_2^2\epsilon^2 L_0}{2},
\end{aligned}
$$

then from upper bound of $\mathbb{E}\left[\left\|v_k - \hat{\nabla} f(x_k)\right\|^2\right]$ (lemma E.3), we have $\mathbb{E}\left[\left\|v_k - \hat{\nabla} f(x_k)\right\|\right] \leq 4L_0c_2\epsilon + 3\sqrt{d}\mu L_0 + \frac{2\sigma_0}{\sqrt{B}} + \sum_{l=\hat{k}}^{k-1}\left(\frac{1}{\sqrt{b}}\left(4L_1c_2\epsilon + 3\sqrt{d}\mu L_1\right) + \frac{2L_1\mu\sqrt{d} + 2\sigma_1}{\sqrt{B}}\right)\left\|\nabla f(x_l)\right\|.$

$$
\begin{aligned}
& \mathbb{E}[f\left(x_{k+1}\right) - f\left(x_k\right)] \\
\leq & -\left(\left(4L_1c_2\epsilon + 3\sqrt{d}\mu L_1\right) + \frac{2L_1\mu\sqrt{d} + 2\sigma_1}{\sqrt{B}}\right)\left\|\nabla f\left(x_k\right)\right\| + \frac{17L_0c_2^2\epsilon^2}{2} + 7L_0c_2\epsilon\sqrt{d}\mu + \frac{2c_2\epsilon\sigma_0}{\sqrt{B}} \\
& + 2c_2\epsilon\sum_{l=\hat{k}}^{k-1}\left(4L_1c_2\epsilon + 3\sqrt{d}\mu L_1 + \frac{2L_1\mu\sqrt{d} + 2\sigma_1}{\sqrt{B}}\right)\left\|\nabla f(x_l)\right\|,
\end{aligned}
$$

sum it from $k = \hat{k}$ to $\hat{k} + q$, we obtain:

$$
\begin{aligned}
& \mathbb{E}[f\left(x_{q+\hat{k}}\right) - f\left(x_{\hat{k}}\right)] \\
\leq & -\left(c_2\epsilon - \frac{L_1c_2^2\epsilon^2}{2} - L_1c_2\epsilon\sqrt{d}\mu\right)\sum_{k=\hat{k}}^{q+\hat{k}-1}\left\|\nabla f\left(x_k\right)\right\| + q\left(\frac{17L_0c_2^2\epsilon^2}{2} + 7L_0c_2\epsilon\sqrt{d}\mu + \frac{2c_2\epsilon\sigma_0}{\sqrt{B}}\right) \\
& + 2c_2\epsilon\sum_{k=\hat{k}}^{\hat{k}+q-1}\sum_{l=\hat{k}}^{k}\left(\frac{1}{\sqrt{b}}\left(4L_1c_2\epsilon + 3\sqrt{d}\mu L_1\right) + \frac{2L_1\mu\sqrt{d} + 2\sigma_1}{\sqrt{B}}\right)\left\|\nabla f(x_l)\right\| \\
\overset{a}{\leq} & -c_2\epsilon\left(1 - L_1c_2\epsilon(\frac{1}{2} + 8\sqrt{q}) - L_1\sqrt{d}\mu(1 + 6\sqrt{q}) - \frac{2q}{\sqrt{B}}(2L_1\mu\sqrt{d} + 2\sigma_1)\right)\sum_{k=\hat{k}}^{q+\hat{k}-1}\left\|\nabla f\left(x_l\right)\right\| \\
& + qc_2\epsilon\left(\frac{17L_0c_2\epsilon}{2} + 7L_0\sqrt{d}\mu + \frac{2\sigma_0}{\sqrt{B}}\right) \\
\overset{b}{\leq} & -\frac{5c_2\epsilon}{8}\sum_{k=\hat{k}}^{q+\hat{k}-1}\left\|\nabla f\left(x_l\right)\right\| + \frac{qc_2\epsilon^2}{4} + 2qc_2\epsilon\frac{\sigma_0}{\sqrt{B}} \\
\overset{c}{\leq} & -\frac{qc_2\epsilon^2}{4},
\end{aligned}
\tag{22}
$$

step(a) holds due to the observation that $\sum_{k=\hat{k}}^{\hat{k}+q}\sum_{l=\hat{k}}^{k}(c_2\epsilon L_1 + 2d\mu L_1)\left\|\nabla f(x_l)\right\| \leq q\sum_{l=\hat{k}}^{q}\left\|\nabla f\left(x_l\right)\right\|$

step(b) holds because we choose $q = \epsilon^{-1}$, then choose $c_2 \leq \min\{\frac{1}{72L_1}, \frac{1}{68L_0}\}$ to let $L_1c_2\epsilon(\frac{1}{2} + 8\sqrt{q}) \leq 9L_1c_2 \leq \frac{1}{8}$ and $\frac{17c_2\epsilon L_0}{2} \leq \frac{\epsilon}{8}$ and $\frac{qc_2\epsilon(2L_1\mu d + 2\sigma_1)}{\sqrt{B}} \leq \frac{\epsilon}{8}, \mu \leq \min\{\frac{\epsilon}{56dL_0}, \frac{1}{56L_1\sqrt{d}\epsilon^{-0.5}}\}$to

let $L_1\sqrt{d}\mu(1 + 6\sqrt{q}) \leq 7L_1\sqrt{d}\mu\epsilon^{-0.5} \leq \frac{1}{8}$ and $7d\mu L_0 \leq \frac{\epsilon}{8}$, choose $B \geq \mathcal{O}(\sigma_1^2\epsilon^{-2})$ to let $\frac{2q}{\sqrt{B}}(2L_1\mu\sqrt{d} + 2\sigma_1) \leq \frac{1}{8}$

step(c)holds due to $\|\nabla f(x_k)\| \geq \epsilon$ otherwise we have find the stationary point, and choose $B \geq \mathcal{O}(\epsilon^{-2}\sigma_0^2)$ to let $2qc_2\epsilon\frac{\sigma_0}{\sqrt{B}} \leq \frac{qc_2\epsilon^2}{8}$

Now, from(21), we know that in the sense of expecation, $f(x)$ descrease at least $\frac{qc_2\epsilon^2}{4}$ in $q$ steps, that is , $\frac{c_2\epsilon}{4}$ every step, since $f(x)$ can deacrease at most $\Delta$, we need at most

$$K = \mathcal{O}(\Delta\epsilon^{-2}\max\{L_1, L_0\}),$$

to find the stationary point, and the total number of oracle calls is

$$\#funtion = dT(b + \frac{B}{q}) = \mathcal{O}(d\epsilon^{-3}\max\{L_1, L_0\}\max\{\sigma_0^2, \sigma_1^2\} + dn\epsilon^{-2}\sigma_0^2).$$

$\square$

## E.2 Convergence analysis of rand estimator

In the following lemma, we disscuss the behavior of variance term $\left\|v_k - \hat{\nabla}f(x_k)\right\|^2$ in finite sum case.

**Lemma E.4** (Variance of finite sum case). *Under assumptions 1 and 2 , for Algorithm 1, let $\mu \leq \frac{1}{dL_1}$ we have*

$$\mathbb{E}\left[\left\|v_k - \bar{\nabla}_S f(x_k)\right\|\right] \leq 6L_0c_2\epsilon + 3d\mu L_0 + \frac{1}{\sqrt{b}}\sum_{l=\hat{k}}^{k}\left(6L_1c_2\epsilon + 3d^2\mu^2L_1^2 + 5\sqrt{\frac{d}{S}}\right)\|\nabla f(x_l)\|.$$

*Proof.* note that $\mathbb{E}[\hat{\nabla}f(x_k; \xi)] = \bar{\nabla}_S f(x_k)$

$$\mathbb{E}\left[\left\|v_{k+1} - \bar{\nabla}_S f(x_k)\right\|^2\right]$$

$$\leq \mathbb{E}[\left\|v_k + \frac{1}{b}\sum_{i=1}^{b}(\bar{\nabla}_S f_i(x_{k+1}) - \bar{\nabla}_S f_i(x_k)) \pm \bar{\nabla}_S f_i(x_k) - \bar{\nabla}_S f(x_{k+1})\right\|^2]$$

$$\overset{a}{\leq} \mathbb{E}\left[\left\|v_k - \bar{\nabla}_S f_i(x_k)\right\|^2\right] + \mathbb{E}\left[\left\|\frac{1}{b}\sum_{i=1}^{b}(\bar{\nabla}_S f_i(x_{k+1}) - \bar{\nabla}_S f_i(x_k)) + \bar{\nabla}_S f(x_k) - \bar{\nabla}_S f(x_{k+1})\right\|^2\right]$$

$$\overset{b}{\leq} \mathbb{E}\left[\left\|v_k - \bar{\nabla}_S f_i(x_k)\right\|^2\right] + \frac{1}{b}\mathbb{E}\left[\left\|\bar{\nabla}_S f_i(x_{k+1}) - \bar{\nabla}_S f_i(x_k)\right\|^2\right]$$

$$\overset{c}{\leq} \mathbb{E}\left[\left\|v_k - \bar{\nabla}_S f_i(x_k)\right\|^2\right]$$

$$+ \frac{1}{b}\left(6\mu^2d^2L_0^2 + 9d^2L_1^2\mu^2\|\nabla f(x_1)\|^2 + 3L_0^2(4 + \frac{d}{S})\|x_1 - x_2\|^2 + (12L_0^2 + \frac{3}{2} + \frac{3dL_1^2}{S})\|\nabla f(x_1)\|^2\|x_1 - x_2\|^2\right),$$

where step (a) holds due to $\mathbb{E}[\frac{1}{b}\sum_{i=1}^{b}(\hat{\nabla}f_i(x_{k+1}) - \hat{\nabla}f_i(x_k)) + \hat{\nabla}f(x_k) - \hat{\nabla}f(x_{k+1})] = 0$ and lemma B.4.

step(b) holds due to lemma B.5.($\mathbb{E}\left[\left\|\frac{1}{b}\sum_{i=1}^{b}x_i - x\right\|^2\right] \leq \frac{\mathbb{E}[\|x_i\|^2]}{b}$).

step(c) holds due to lemma D.5 to upper bound $\mathbb{E}\left[\left\|\bar{\nabla}_S f_i(x_{k+1}) - \bar{\nabla}_S f_i(x_k)\right\|^2\right]$.

denote $\hat{k} = \lfloor k/q \rfloor q$, note that (i)$\|x_{k+1} - x_k\| \leq c_2\epsilon$ due to the choice of $\eta_k$, (ii) for Algorithm 1, in finite sum case, when $k = \hat{k}$, we use all $n$ samples to compute the gradient, which means

$v_{\hat{k}} = \bar{\nabla}_S f(x_k)$, so $\mathbb{E}\left[\left\|v_{\hat{k}} - \bar{\nabla}_S f(x_k)\right\|^2\right] = \mathbb{E}\left[\left\|\bar{\nabla}_S f(x_k) - \bar{\nabla}_S f(x_k)\right\|^2\right] = 0$, we obtain

$$\mathbb{E}\left[\left\|v_k - \bar{\nabla}_S f(x)\right\|^2\right]$$

$$\leq \frac{1}{b}\sum_{l=\hat{k}}^{k}\left(6\mu^2 d^2 L_0^2 + 9d^2 L_1^2 \mu^2 \left\|\nabla f(x_1)\right\|^2 + 3L_0^2(4 + \frac{d}{S})\left\|x_1 - x_2\right\|^2 + (12L_0^2 + \frac{3}{2} + \frac{3dL_1^2}{S})\left\|\nabla f(x_1)\right\|^2 \left\|x_1 - x_2\right\|^2\right)$$

$$+ \underbrace{\mathbb{E}\left[\left\|v_{\hat{k}} - \bar{\nabla}_S f(x_{\hat{k}})\right\|^2\right]}_{=0(\text{finite sum case})}$$

$$\overset{a}{\leq} \frac{q}{b}\left(15L_0^2 c_2^2 \epsilon^2 + 6L_0^2 d^2 \mu^2\right) + \frac{1}{b}\sum_{l=\hat{k}}^{k}\left((3L_1^2 + \frac{3}{2} + 12L_0^2)c_2^2 \epsilon^2 + 9d^2 \mu^2 L_1^2\right)\left\|\nabla f(x_l)\right\|^2, \qquad (23)$$

where step (a) holds because we choose $S \geq d$ and let $\mu \leq \frac{1}{dL_1}$.

Let $q = b$, use the fact that $\sqrt{\sum \|x_i\|^2} \leq \sum \|x_i\|$ when every $x_i \geq 0$, and $\mathbb{E}[\|x\|] \leq \sqrt{\mathbb{E}[\|x\|^2]}$, we have

$$\mathbb{E}\left[\left\|v_k - \bar{\nabla}_S f(x)\right\|\right] \leq \sqrt{15}L_0 c_2 \epsilon + \sqrt{6}\mu d L_0 + \frac{1}{\sqrt{b}}\sum_{l=\hat{k}}^{k}\left((\sqrt{3}L_1 + \sqrt{\frac{3}{2}} + 2\sqrt{3}L_0)c_2 \epsilon + 3d^2 \mu^2 L_1^2\right)\left\|\nabla f(x_l)\right\|.$$

To improve readability, we perform some scaling on square root terms:

$$\mathbb{E}\left[\left\|v_k - \bar{\nabla}_S f(x)\right\|\right] \leq 4L_0 c_2 \epsilon + 3\mu d L_0 + \frac{1}{\sqrt{b}}\sum_{l=\hat{k}}^{k}\left((2L_1 + 2 + 4L_0)c_2 \epsilon + 3L_1 \mu d\right)\left\|\nabla f(x_l)\right\|$$

$$. \hspace{13cm} \square$$

In the following lemma, we disscuss the behavior of variance term $\left\|v_k - \bar{\nabla}_S f(x_k)\right\|^2$ in Expectation case,

**Lemma E.5** (Variance of Expectation case). *Under assumptions 1 and 2 , for Algorithm 1, let $q = b$, $\mu \leq \frac{1}{dL_1}$, we have:*

$$\mathbb{E}\left[\left\|v_k - \bar{\nabla}_S f(x_k)\right\|\right] \leq 6L_0 c_2 \epsilon + 4d\mu L_0 + \frac{3\sigma_0}{\sqrt{B}}$$

$$+ \sum_{l=\hat{k}}^{k}\left(\frac{1}{\sqrt{b}}\left((2L_1 + 2 + 4L_0)c_2 \epsilon + 3L_1 \mu d\right) + \frac{3(3 + \sigma_1)}{\sqrt{B}}\right)\left\|\nabla f(x_l)\right\|,$$

*Proof.* Similar to the previous theorem, but in Expectation case, $\left\|v_{\hat{k}} - \bar{\nabla}_S f(x_{\hat{k}})\right\|^2 \neq 0$, we start from (23) and obtain:

$$\mathbb{E}\left[\left\|v_k - \bar{\nabla}_S f(x_k)\right\|^2\right] \hspace{9cm} (24)$$

$$\leq \frac{q}{b}\left(15L_0^2 c_2^2 \epsilon^2 + 6L_0^2 d^2 \mu^2\right) + \frac{1}{b}\sum_{l=\hat{k}}^{k}\left((3L_1^2 + \frac{3}{2} + 12L_0^2)c_2^2 \epsilon^2 + 9d^2 \mu^2 L_1^2\right)\left\|\nabla f(x_l)\right\|^2 + \mathbb{E}\left[\left\|v_{\hat{k}} - \bar{\nabla}_S f(x_{\hat{k}})\right\|^2\right]$$

$$\hspace{14cm} (25)$$

$$\leq \frac{q}{b}\left(15L_0^2 c_2^2 \epsilon^2 + 6L_0^2 d^2 \mu^2\right) + \frac{1}{b}\sum_{l=\hat{k}}^{k}\left((3L_1^2 + \frac{3}{2} + 12L_0^2)c_2^2 \epsilon^2 + 9d^2 \mu^2 L_1^2\right)\left\|\nabla f(x_l)\right\|^2 \hspace{1cm} (26)$$

$$+ \frac{1}{B}\left(6(\mu^2 d^2 L_1^2 + \frac{2d}{S} + \sigma_1^2)\left\|\nabla f(x_{\hat{k}})\right\|^2 + 6\mu^2 d^2 L_0^2 + 6\sigma_0^2\right), \hspace{3cm} (27)$$

where the second inequality holds due to $\mathbb{E}\left\|\nabla_S f(x;\xi) - \nabla_S f(x)\right\|^2 \leq 6(\mu^2 d^2 L_1^2 + \frac{2d}{S} + \sigma_1^2)\left\|\nabla f(x)\right\|^2 + 6\mu^2 d^2 L_0^2 + 6\sigma_0^2$ from lemma D.6.

Let $q = b$, $\mu \leq \frac{1}{dL_1}$, $S \geq d$, use the fact that $\sqrt{\sum \|x_i\|^2} \leq \sum \|x_i\|$ when every $x_i \geq 0$, and $\mathbb{E}[\|x\|] \leq \sqrt{\mathbb{E}[\|x\|^2]}$, we have

$$\mathbb{E}\left[\left\|v_k - \bar{\nabla}_S f(x_k)\right\|^2\right]$$

$$\leq 15 L_0^2 c_2^2 \epsilon^2 + 6 L_0^2 d^2 \mu^2 + \frac{6 L_0^2 \mu^2 d^2}{B} + \frac{6\sigma_0^2}{B}$$

$$+ \sum_{l=\hat{k}}^{k} \left(\frac{1}{b}\left((3L_1^2 + \frac{3}{2} + 12 L_0^2)c_2^2 \epsilon^2 + 9 d^2 \mu^2 L_1^2\right) + \frac{6(3+\sigma_1^2)}{B}\right)\left\|\nabla f(x_l)\right\|^2,$$

now, use the fact that $\sqrt{\sum \|x_i\|^2} \leq \sum \sqrt{x_i}$ when every $x_i \geq 0$, and $\mathbb{E}[\|x\|] \leq \sqrt{\mathbb{E}[\|x\|^2]}$, and $B \gg 1$, $S \geq 1$ we have

$$\mathbb{E}\left[\left\|v_k - \bar{\nabla}_S f(x_k)\right\|\right] \leq 6 L_0 c_2 \epsilon + 4 d\mu L_0 + \frac{3\sigma_0}{\sqrt{B}}$$

$$+ \sum_{l=\hat{k}}^{k} \left(\frac{1}{\sqrt{b}}\left((2L_1 + 2 + 4 L_0)c_2 \epsilon + 3 L_1 \mu d\right) + \frac{3(3+\sigma_1)}{\sqrt{B}}\right)\left\|\nabla f(x_l)\right\|,$$

$\square$

**Theorem E.3** (Restatement of Theorem 3 (rand estimator in finite sum case)). *For Algorithm 1, under assumptions 1 and 2, let $c_2 \leq \min\{\frac{1}{8(3L_1+2+4L_0)}, \frac{1}{36 L_0}\}$, choose $\eta_k = \frac{c_2\epsilon}{\|v_k\|}, q = b = \sqrt{n}, B = n$, $\mu \leq \min\{\frac{\epsilon}{40 d L_0}, \frac{1}{20 L_1 d}\}$, we obtain:*

$$\mathbb{E}[f\left(x_{q+\hat{k}}\right) - f\left(x_{\hat{k}}\right)] \leq -\frac{3 q c_2 \epsilon^2}{8}.$$

*We state that, in expectation, the function value of $f$ decreases by an average of $\min\{\frac{1}{50 L_1}, \frac{1}{20 L_0}\}$ in, since $f(x)$ per iteration. Since $f$ can deacrease at most $\Delta$, we need at most*

$$K = \mathcal{O}(\Delta\epsilon^{-2}\max\{L_1, L_0\}),$$

*in expectation to find the stationary point, and the total number of e function query is*

$$\#function\ query = dT(b + \frac{B}{q}) = \mathcal{O}(d\epsilon^{-2}\sqrt{n}\max\{L_1, L_0\} + dn).$$

*Proof.* From the conclusion of lemma E.1, we have

$$f\left(x_{k+1}\right) \leq f\left(x_k\right) - \left(c_2\epsilon - \frac{c_2^2\epsilon^2}{2}L_1\right)\left\|\nabla f\left(x_k\right)\right\| + 2 c_2\epsilon\left\|v_k - \nabla f\left(x_k\right)\right\| + \frac{c_2^2\epsilon^2 L_0}{2}$$

$$\leq f\left(x_k\right) - \left(c_2\epsilon - \frac{c_2^2\epsilon^2}{2}L_1\right)\left\|\nabla f\left(x_k\right)\right\| + 2 c_2\epsilon\left(\left\|\bar{\nabla}_S f(x) - \nabla f(x)\right\| + \left\|v_k - \bar{\nabla}_S f(x)\right\|\right) + \frac{c_2^2\epsilon^2 L_0}{2},$$

then from upper bound $\left\|\bar{\nabla}_S f(x) - \nabla f(x)\right\|^2 \leq \mathbb{E}\left[\left\|\bar{\nabla}_S f(x) - \nabla f(x)\right\|^2\right] \leq (\mu^2 d^2 L_1^2 + \frac{2d}{S})\left\|\nabla f(x)\right\|^2 + \mu^2 d^2 L_0^2$ (lemma D.4), upper bound $\mathbb{E}\left[\left\|v_k - \bar{\nabla}_S f(x_k)\right\|\right] \leq 6 L_0 c_2 \epsilon + 3 d\mu L_0 + \frac{1}{b}\sum_{l=\hat{k}}^{k}\left(6 L_1 c_2 \epsilon + 3 d^2 \mu^2 L_1^2 + 5\sqrt{\frac{d}{S}}\right)\left\|\nabla f(x_l)\right\|,$

$$\mathbb{E}[f\left(x_{k+1}\right) - f\left(x_k\right)]$$

$$\leq -\left(c_2\epsilon - \frac{L_1 c_2^2\epsilon^2}{2} - 2 L_1 c_2 \epsilon d\mu - 4 c_2\epsilon\sqrt{\frac{d}{S}}\right)\left\|\nabla f\left(x_k\right)\right\|$$

$$+ \frac{9 L_0 c_2^2\epsilon^2}{2} + 4 L_0 c_2 \epsilon d\mu + \frac{2 c_2\epsilon}{\sqrt{b}}\sum_{l=\hat{k}}^{k}\left((2L_1 + 2 + 4 L_0)c_2 \epsilon + 3 L_1 \mu d\right)\left\|\nabla f(x_l)\right\|,$$

sum it from $k = \hat{k}$ to $q$, we obtain

$$\mathbb{E}[f\left(x_{q+\hat{k}}\right) - f\left(x_{\hat{k}}\right)]$$

$$\leq - \left(c_2\epsilon - \frac{L_1 c_2^2 \epsilon^2}{2} - 2L_1 c_2 \epsilon d\mu - 4c_2\epsilon\sqrt{\frac{d}{S}}\right) \sum_{k=\hat{k}}^{q+\hat{k}-1} \|\nabla f\left(x_k\right)\| + q\left(\frac{9L_0 c_2^2 \epsilon^2}{2} + 4L_0 c_2 \epsilon d\mu\right)$$

$$+ \frac{2c_2\epsilon}{\sqrt{b}} \sum_{k=\hat{k}}^{\hat{k}+q-1} \sum_{l=\hat{k}}^{k} ((2L_1 + 2 + 4L_0)c_2\epsilon + 3L_1\mu d)\|\nabla f(x_l)\|$$

$$\overset{a}{\leq} - c_2\epsilon \left(1 - c_2\epsilon(\frac{L_1}{2} + \sqrt{q}(2L_1 + 2 + 4L_0)) - (2 + 6\sqrt{q})L_1 d\mu - 4\sqrt{\frac{d}{S}}\right) \sum_{k=\hat{k}}^{q+\hat{k}-1} \|\nabla f\left(x_l\right)\|$$

$$+ qc_2\epsilon \left(\frac{9L_0 c_2\epsilon}{2} + 4L_0 d\mu\right)$$

$$\overset{b}{\leq} - c_2\epsilon \left(1 - c_2(3L_1 + 2 + 4L_0) - (2 + 6\sqrt{q})L_1 d\mu - 4\sqrt{\frac{d}{S}}\right) \sum_{k=\hat{k}}^{q+\hat{k}-1} \|\nabla f\left(x_l\right)\| + qc_2\epsilon \left(\frac{9L_0 c_2\epsilon}{2} + 4L_0 d\mu\right)$$

$$\overset{c}{\leq} - \frac{5c_2\epsilon}{8} \sum_{k=\hat{k}}^{q+\hat{k}-1} \|\nabla f\left(x_l\right)\| + \frac{c_2\epsilon^2}{4}$$

$$\overset{d}{\leq} - \frac{3qc_2\epsilon^2}{8},$$

where step (a) holds due to the observation that $\sum_{k=\hat{k}}^{\hat{k}+q} \sum_{l=\hat{k}}^{k}(c_2\epsilon L_1 + 2d\mu L_1)\|\nabla f(x_l)\| \leq q\sum_{l=\hat{k}}^{q}\|\nabla f\left(x_l\right)\|$ and choose $b = q = \sqrt{n}$.

step(b)holds because we suppose $\epsilon \leq \frac{1}{2n^{\frac{1}{4}}}$, and $n \geq 1$ so that $\epsilon\sqrt{q} = \epsilon n^{\frac{1}{4}} \leq 1$.

step(c) holds because we choose $c_2 \leq \min\{\frac{1}{8(3L_1+2+4L_0)}, \frac{1}{36L_0}\}$ to let $c_2(3L_1 + 2 + 4L_0) \leq \frac{1}{8}$ and $\frac{9c_2\epsilon L_0}{2} \leq \frac{\epsilon}{8}$, and $\mu \leq \min\{\frac{\epsilon}{32dL_0}, \frac{1}{8(2+6n^{\frac{1}{4}})L_1 d}\}$ to let $8n^{\frac{1}{4}}L_1 d\mu \leq \frac{1}{8}$ and $4d\mu L_0 \leq \frac{\epsilon}{8}$, choose $S = \mathcal{O}(d)$ to let $4\sqrt{\frac{d}{S}} \leq \frac{1}{8}$.

step(d) holds dut to $\|\nabla f(x_k)\| \geq \epsilon$ otherwise we have find the stationary point.

Now, from(21), we know that in the sense of expecation, $f(x)$ descrease at least $\frac{qc_2\epsilon^2}{4}$ in $q$ steps, that is , $\min\{\frac{1}{50L_1}, \frac{1}{20L_0}\}$ every step in average, since $f(x)$ can deacrease at most $\Delta$, we need at most

$$K = \mathcal{O}(\Delta\epsilon^{-2}\max\{L_1, L_0\})$$

in expectation to find the stationary point, and the total number of oracle calls is

$$\#funtion = \mathcal{O}(d)K(b + \frac{B}{q}) = \mathcal{O}(d\epsilon^{-2}\sqrt{n}\max\{L_1, L_0\} + dn).$$

$\square$

**Theorem E.4** (Restatement of Theorem 4 (rand estimator in expecation case))**.** *For Algorithm 1, under assumptions 1 and 2, let $c_2 \leq \min\{\frac{1}{8(5L_1+2+4L_0)}, \frac{1}{36L_0}\}$, choose $\eta_k = \frac{c_2\epsilon}{\|v_k\|}$, $\mu \leq \min\{\frac{\epsilon}{40dL_0}, \frac{1}{20L_1 d}\}$, $B \geq \max\{\mathcal{O}(\epsilon^{-2}(3 + \sigma_1)^2), \mathcal{O}(\epsilon^{-2}\sigma_0^2)\}$, we have*

$$\mathbb{E}[f\left(x_{q+\hat{k}}\right) - f\left(x_{\hat{k}}\right)] \leq -\frac{qc_2\epsilon^2}{8}$$

*We state that, in expectation, the function value of $f$ decreases by an average of$\min\{\frac{1}{50L_1}, \frac{1}{20L_0}\}$ in, since $f(x)$ per iteration.Since $f$ can deacrease at most $\Delta$, we need at most*

$$K = \mathcal{O}(\Delta\epsilon^{-2}\max\{L_1, L_0\})$$

*in expectation to find the stationary point, and the total number of e function query is*

$$\#funtion = \mathcal{O}(d)K(b + \frac{B}{q}) = \mathcal{O}(d\epsilon^{-3}\max\{\sigma_1^2, \sigma_0^2\}\max\{L_1, L_0\} + \epsilon^2\max\{\sigma_1^2, \sigma_0^2\}).$$

*Proof.* From the conclusion of lemma E.1, we have

$$f(x_{k+1}) \leq f(x_k) - \left(c_2\epsilon - \frac{c_2^2\epsilon^2}{2}L_1\right)\|\nabla f(x_k)\| + 2c_2\epsilon \|v_k - \nabla f(x_k)\| + \frac{c_2^2\epsilon^2 L_0}{2}$$

$$\leq f(x_k) - \left(c_2\epsilon - \frac{c_2^2\epsilon^2}{2}L_1\right)\|\nabla f(x_k)\| + 2c_2\epsilon\left(\|\bar{\nabla}_S f(x) - \nabla f(x)\| + \|v_k - \bar{\nabla}_S f(x)\|\right) + \frac{c_2^2\epsilon^2 L_0}{2},$$

then from upper bound $\mathbb{E}\left[\|\bar{\nabla}_S f(x) - \nabla f(x)\|^2\right] \leq (\mu^2 d^2 L_1^2 + \frac{2d}{S})\|\nabla f(x)\|^2 + \mu^2 d^2 L_0^2$(lemma D.4), upper bound $\mathbb{E}\left[\|v_k - \bar{\nabla}_S f(x_k)\|\right] \leq 6L_0 c_2\epsilon + 4d\mu L_0 + \frac{3\sigma_0}{\sqrt{B}} + \sum_{l=\hat{k}}^{k}\left(\frac{1}{\sqrt{b}}((2L_1 + 2 + 4L_0)c_2\epsilon + 3L_1\mu d) + \frac{3(3+\sigma_1)}{\sqrt{B}}\right)\|\nabla f(x_l)\|$ (lemma E.5) .

$$\mathbb{E}[f(x_{k+1}) - f(x_k)]$$

$$\leq -\left(c_2\epsilon - \frac{L_1 c_2^2\epsilon^2}{2} - 2L_1 c_2\epsilon d\mu - 4c_2\epsilon\sqrt{\frac{d}{S}}\right)\|\nabla f(x_k)\| + \frac{6c_2\epsilon\sigma_0}{\sqrt{B}}$$

$$+ \frac{13L_0 c_2^2\epsilon^2}{2} + 5L_0 c_2\epsilon d\mu + 2c_2\epsilon\sum_{l=\hat{k}}^{k}\left(\frac{1}{\sqrt{b}}((2L_1 + 2 + 4L_0)c_2\epsilon + 3L_1\mu d) + \frac{3(3+\sigma_1)}{\sqrt{B}}\right)\|\nabla f(x_l)\|,$$

sum it from $k = \hat{k}$ to $q$, we obtain

$$\mathbb{E}[f(x_{q+\hat{k}}) - f(x_{\hat{k}})]$$

$$\leq -\left(c_2\epsilon - \frac{L_1 c_2^2\epsilon^2}{2} - 2L_1 c_2\epsilon d\mu - 4c_2\epsilon\sqrt{\frac{d}{S}}\right)\sum_{k=\hat{k}}^{q+\hat{k}-1}\|\nabla f(x_k)\| + q\left(\frac{13L_0 c_2^2\epsilon^2}{2} + 5L_0 c_2\epsilon d\mu + \frac{3\sigma_0}{\sqrt{B}}\right)$$

$$+ 2c_2\epsilon\sum_{k=\hat{k}}^{\hat{k}+q-1}\sum_{l=\hat{k}}^{k}\left(\frac{1}{\sqrt{b}}((2L_1 + 2 + 4L_0)c_2\epsilon + 3L_1\mu d) + \frac{3(3+\sigma_1)}{\sqrt{B}}\right)\|\nabla f(x_l)\|$$

$$\overset{a}{\leq} -c_2\epsilon\left(1 - c_2\epsilon(\frac{L_1}{2} + \sqrt{q}(4L_1 + 4 + 8L_0)) - (2 + 6\sqrt{q})L_1 d\mu - 4\sqrt{\frac{d}{S}} - \frac{6q(3+\sigma_1)}{\sqrt{B}}\right)\sum_{k=\hat{k}}^{q+\hat{k}-1}\|\nabla f(x_l)\|$$

$$+ qc_2\epsilon\left(\frac{13L_0 c_2^2\epsilon^2}{2} + 5L_0 c_2\epsilon d\mu + \frac{6\sigma_0}{\sqrt{B}}\right)$$

$$\overset{b}{\leq} -c_2\epsilon\left(1 - c_2(5L_1 + 2 + 4L_0) - (2 + 6\epsilon^{-0.5})L_1 d\mu - 4\sqrt{\frac{d}{S}} - \frac{6q(3+\sigma_1)}{\sqrt{B}}\right)\sum_{k=\hat{k}}^{q+\hat{k}-1}\|\nabla f(x_l)\|$$

$$+ qc_2\epsilon\left(\frac{13L_0 c_2^2\epsilon^2}{2} + 5L_0 c_2\epsilon d\mu + \frac{6\sigma_0}{\sqrt{B}}\right)$$

$$\overset{c}{\leq} -\frac{1 c_2\epsilon}{2}\sum_{k=\hat{k}}^{q+\hat{k}-1}\|\nabla f(x_l)\| + \frac{3qc_2\epsilon^2}{8}$$

$$\overset{d}{\leq} -\frac{qc_2\epsilon^2}{8},$$

where step (a) holds due to the observation that $\sum_{k=\hat{k}}^{\hat{k}+q}\sum_{l=\hat{k}}^{k}(c_2\epsilon L_1 + 2d\mu L_1)\|\nabla f(x_l)\| \leq q\sum_{l=\hat{k}}^{q}\|\nabla f(x_l)\|$ and

step(b) holds because we let $b = q = \epsilon^{-1}$ , $S = \mathcal{O}(d)$ such that $4\sqrt{\frac{d}{S}} \leq \frac{1}{8}$,

step(c) holds because we choose $c_2 \leq \min\{\frac{1}{8(5L_1+2+4L_0)}, \frac{1}{36L_0}\}$ to let $c_2(5L_1 + 2 + 4L_0) \leq \frac{1}{8}$ and $\frac{9c_2\epsilon L_0}{2} \leq \frac{\epsilon}{8}$, $\mu \leq \min\{\frac{\epsilon}{32dL_0}, \frac{1}{8(2+6\epsilon^{-0.5})L_1 d}\}$ to let $(2 + 6\epsilon^{-0.5}L_1 d\mu) \leq \frac{1}{8}$ and $4d\mu L_0 \leq \frac{\epsilon}{8}$, choose $S = \mathcal{O}(d)$ to let $4\sqrt{\frac{d}{S}} \leq \frac{1}{8}$, $B \geq \max\{\mathcal{O}(\epsilon^{-2}(3 + \sigma_1)^2), \mathcal{O}(\epsilon^{-2}\sigma_0^2)\}$ to let $\frac{6q(3+\sigma_1)}{\sqrt{B}} \leq \frac{1}{8}$, $\frac{6\sigma_0}{\sqrt{B}} \leq \frac{\epsilon}{8}$.

step(d) holds dut to $\|\nabla f(x_k)\| \geq \epsilon$ otherwise we have find the stationary point.

Now, from(21), we know that in the sense of expecation, $f(x)$ descrease at least $\frac{qc_2\epsilon^2}{4}$ in $q$ steps, that is , $\frac{c_2\epsilon}{4}$ every step in average, since $f(x)$ can deacrease at most $\Delta$, we need at most

$$K = \mathcal{O}(\Delta\epsilon^{-2}\max\{L_1, L_0\}),$$

in expectation to find the stationary point, and the total number of oracle calls is

$$\#funtion = \mathcal{O}(d)K(b + \frac{B}{q}) = \mathcal{O}(d\epsilon^{-3}\max\{\sigma_1^2, \sigma_0^2\}\max\{L_1, L_0\} + \epsilon^2\max\{\sigma_1^2, \sigma_0^2\}).$$

$\square$

