# OpenReview forum: "Variance-Reduced Normalized Zeroth Order Method for Generalized-Smooth Non-Convex Optimization"
_ICLR.cc/2025/Conference — Submitted to ICLR 2025_

### Official Review · Reviewer_XunU · 2024-11-01

**Soundness:** 2
**Presentation:** 1
**Contribution:** 2
**Rating:** 3
**Confidence:** 5

**Summary:**

The paper studies zeroth-order optimization under the generalized $(L_0,L_1)$-smoothness condition. Previous works under this assumption focus on first-order methods, where clipped SGD achieves $\epsilon^{-4}$ rate, the same rate as SGD under $L$-smoothness condition, and clipped SPIDER achieves $\epsilon^{-3}$ rate, the same rate as SPIDER under average or individual smoothness condition. Here, the authors replace gradients in normalized SPIDER with their zeroth-order estimators and achieve $d\epsilon^{-3}$ rate, which is the first result for zeroth-order optimization under generalized smoothness condition.

**Strengths:**

The topic on zeroth-order optimization is important in the optimization and machine learning community, due to many interesting applications such as black-box attack and reinforcement learning. While most previous works on zeroth-order methods assume the smoothness condition, there are many settings where this assumption fails to hold. Generalized smoothness condition provides one relaxation and fits several real-world applications. Research on generalized smoothness condition sets good examples on efforts to close the gap between theoretical setups and pratical applications.

**Weaknesses:**

1. There are so many typos and grammar issues in the paper. This was really annoying when I was reading the paper. I am not sure whether this comes from that this paper was written in a rush or are there any other reasons? If the paper was written in a rush, then I would reasonably question the correctness of the theoretical proofs. Here are some issues (I could not list all of them as there are so many). line21 in the abstract: complexes; line86-93: what do you mean by "we both analyze, we both use"; line98: can as effective as; line123 in table 2: estimator denotes represent the number; line194, 239: can't access to; line265: should be introduced; line308 in Algo. 1: defied in (5) and (4); line364: constans; line376: itration; line382, 408, 420: funtion; line408, 420: number of e function query; line 524: coordand rand. Some weird spacing: line40, distribution.In; line 42; line 162. Grammar: You cannot just list a sequence of sentences in English, e.g., "sentence A, sentence B, sentence C". They should be connected using conjunctions (and, but), relative clauses (which, where, that), or transitional words or phrases (however, moreover). This happens several times in the paper, e.g., line 76-79, 187-199, 202-203, 284-286, 347-354, 364-376. I completely understand the strict timeline for submission, where typos and grammar problems are unavoidable. However, such many issues are unaccepted. It disrupts readability and affects clarity of the paper. This is also unfair for other submissions where the authors spend lots of time on polishing the paper. As a reviewer, it is also frustrating to evalute papers that are far from publication standard.

2. Some classical results on zeroth-order methods are missing. $\mathbb{E}\bar \nabla f=\nabla_\mu f$ was first proved in [1] as far as I know. Classical analysis of zeorth-order methods were provided in [2] and [3]. [4] also studied variance reduced zeroth-order method, and [5] discussed the lower-bound.

[1] Online convex optimization in the bandit setting: Gradient descent without a gradient. SODA 2005.

[2] Random gradient-free minimization of convex functions. Nesterov and Spokoiny.

[3] An optimal algorithm for bandit and zero-order convex optimization with two-point feedback. JMLR, 2017.

[4] Zeroth-order stochastic variance reduction for nonconvex optimization. NeurIPS, 2018.

[5] Optimal rates for zero-order convex optimization: The power of two function evaluations. IEEE, 2015.

3. The authors claim their complexity on $\epsilon$ and $d$ is optimal. However, there is no discussion on the lower-bound in the paper under Assumptions 1-3 for zeroth-order methods. Usually, the complexity of zeroth-order methods is $d$-times worse compared to first-order methods (see lower-bounds in [5] above), but that is for $L$-smoothess setting. The author should discuss why the same lower-bound applies to the generalized smoothness setting. Before that, it is only valid to claim the complexity being best-known upper-bound instead of the optimal.

4. Another option for Coord estimator is to only sample a subset of all $d$ coordinates instead of computing every coordinate, as the current one is computational challenge for problem with large dimensions.

5. Although the paper is indeed the first one on zeroth-order optimization with generalized smoothness condition, I am not surprised by any of its results. All the proof techniques exist before, and there are no fundamental difference by combining all related works. First, existing work already show $\epsilon^{-3}$ rate using SPIDER under the generalized smoothness condition. Second, extensive literature provides examples how to extend from first-order methods to zeroth-order methods. The computation involved in Lemmas 1-3 is also standard, and there is nothing special. I can only rate the novelty as marginal.

6. It is unclear why the current experiments verify the effectiveness of the proposed method. I am not sure why one has to apply zeroth-order methods on applications (10) and (9), where gradients are accessible and not hard to compute. These might be good examples for $(L_0, L_1)$-smoothness, but not for why zeroth-order methods should be applied. Why not just use first-order methods on the two applications? Zeroth-order methods are useful when gradient is not possible or hard to obtain, and the current experiments do not belong to this class.

7. In the theoretical results, the complexity of the proposed zeroth-order method is $d\epsilon^{-3}$, while that of first-order methods is $\epsilon^{-3}$. This means zeroth-order method is $d$-times worse compared to first-order methods. Then why is it the case that in Figures 1(c) and 1(d), there is not much difference between zeroth- and first-order methods? What is the sample complexity here in the figure? Is it queries on $f$, or queries on $|S|$ number of $f$?

**Questions:**

See weaknesses.

---

> ### Author Response · Authors · 2024-12-03
>
> Thank you for your valuable feedback. We will incorporate your suggestions to further improve our paper.

---

### Official Review · Reviewer_sY4G · 2024-11-02

**Soundness:** 2
**Presentation:** 2
**Contribution:** 2
**Rating:** 3
**Confidence:** 3

**Summary:**

The paper studies zeros-order methods for stochastic non-convex $(L_0, L_1)$-smooth optimization. The zeroth-order version of SPIDER, namely ZO-normalized-SPIDER, is presented using rand and coord estimators of gradients. New convergence rates are derived for finite-sum and expectation cases for non-convex $(L_0, L_1)$-smooth optimization.  In my opinion, the novelty of the paper is limited. There are two ingredients to the proof:
- in bounding the gradient estimation (rand, coord) error. This part is simple and done by applying previous results and Assumption 1 directly.
- in bounding $\mathbb{E}[\|v_k - \hat{\nabla} f(x, \xi)\|^2]$ in $(L_0, L_1)$-smooth case, which is already done in Chen et al 2023.

**Strengths:**

- Based on the literature provided in the paper, this is the first result on zeroth-order methods for stochastic $(L_0, L_1)$-smooth nonconvex optimization.
- I belive the results are correct with minor errors.

**Weaknesses:**

- The paper is poorly written; there are many typos, and some sentences are confusing and unfinished. The manuscript should be polished and revisited.
-  Assumption A2(3) is a strong assumption that requires the $(L_0, L_1)$-smooth property to be satisfied for every sample.
- The paper does not present and compare with the results on convergence of the SPIDER method for $\alpha$-symmetric functions provided in Chen et al 2023; they also used normalization, and the analysis in their work has the same challenges and tricks used in this paper. ZO-normalized-SPIDER is exactly Algorithm 2 (Spider). In Technical Novelty paragraph in Chen et al 2023, the same problem of bounding $\mathbb{E}[\|v_k - \nabla f(x_k, \xi)\|^2]$ is solved.

Please address the following concerns:
- line 048: A key observation where? This sentence is not complete.
- lines 194-195 the sentence there is incomplete.
- the error bound in Lemmas 1, 4 holds only for $\mu$ is bounded by some constant, perhaps $\frac{1}{L_1}$.
- sentence on lines 392-393 is not clear.
Typos:
- line  041, these problems
- line 042, space between notable.
- lines 047-048, critical applications such as LSTM. Perhaps you want to say critical application such as training of LSTM models.
- line  098, it should be  zeroth-order methods
- line 10,1 it should be convergence instead of converge
- lines 123-124,
>denotes represent

 perhaps one of these is a typo.
- lines 1628-1629  dut should be due
- line 230, approximate should be approximation

**Questions:**

Comments:
-  the definition (3) is not the original definition of L_0, L_1 smooth function. Perhaps the reference to the works where this definition was introduced and studied (see Section 2 in Chen et al 2023) should be included. Vectors x, x' were never defined.

> SGD can't be directly applied to $(L_0, L_1)$ case
-It is worth mentioning that SGD, in fact, converges. In the original work by Zhang et al 2019 the convergence of SGD is provided in Theorem 8, but with a slower rate and additional assumption.

- the last sentence in line 218 is misleading because Lemma 3 does not show Lipschitz continuity not for $\bar{\nabla}_S f(x_1, \xi)$ neither for $\mathbb{E}[\bar{\nabla}_S f(x_1, \xi)]$.

Questions:
- I am also confused by the use of rand and cord in finite sum case. In Algorithm 1, is it required to use a full batch at each iteration to compute a coord estimator? What is $\xi$ in the finite-sum case in Algorithm  1 (ZO-Normalized-SPIDER)? You didn't define it.
-lines 043 044:  faster than what?
>SGD,  variance reduced methods   ... which have demonstrated faster convergence
- line 184 should it be $\frac{d}{\mu}$ instead of $\frac{n}{\mu}$?
- line 203: ...the error... the error of what?

**Details Of Ethics Concerns:**

I don't think paper has any ethics concerns

---

> ### Author Response · Authors · 2024-12-03
>
> Thank you for your valuable feedback. We will incorporate your suggestions to further improve our paper.

---

### Official Review · Reviewer_1T35 · 2024-11-02

**Soundness:** 2
**Presentation:** 2
**Contribution:** 2
**Rating:** 5
**Confidence:** 3

**Summary:**

Authors propose gradient-free methods for non-convex general smooth stochastic optimization problems. As particular case they consider sum-type problem structure.
BTW It seems that the Literature part could be completed by
https://arxiv.org/pdf/2409.14989
https://arxiv.org/pdf/2410.10800

**Strengths:**

I consider the paper contains quite enough mathematics and gives a positive answer for this question:
"Can zeroth-order methods solve generalized (L0, L1)-smooth non-convex problems as efficiently as solving traditional smooth non-convex problems? In particular, what convergence rates can be achieved?" Also authors demonstrate their results by numerical experiments.

**Weaknesses:**

1) rand estimator seems to be not optimal one, e.g. see Shamir's (O. Shamir. An optimal algorithm for bandit and zero-order convex optimization with two-point feedback. Journal of Machine Learning Research, 18(1):1703–1713, 2017.) or Polyak-Tsybakov ones (if you have high-order smoothness) https://arxiv.org/pdf/2306.02159
2) I don't understand how $L_0$ and $L_1$ could be compared in $\max\{\cdot\}$ in formulas. They have principally different physical dimension.
3) There are no high-probability deviations bounds in the paper
4) The only improvement from my point of view is gradient-free generalization of full (stochastic gradient procedures), but nowadays this is quite a routine procedure.

**Questions:**

What is the main contribution from the mathematical point of view (indicate something that was not done by analogue)?

---

> ### Author Response · Authors · 2024-12-03
>
> Thank you for your valuable feedback. We will incorporate your suggestions to further improve our paper.

---

### Official Review · Reviewer_8hDH · 2024-11-04

**Soundness:** 2
**Presentation:** 2
**Contribution:** 2
**Rating:** 3
**Confidence:** 3

**Summary:**

This paper provides variance reduction methods for zeroth-order non-convex optimization under generalized smoothness, including two types of gradient estimators and two cases of problem definitions. The methods achieve an iteration complexity of $\mathcal{O}(\Delta d\epsilon^{-3}\max\\{L_0,L_1\\})$ for finding an $\epsilon$-stationary point.

**Strengths:**

This paper analyzes zeroth-order variance reduction methods under the setting of $(L_0,L_1)$-smoothness, which, to the best of my knowledge, has not been previously investigated.

**Weaknesses:**

My primary concern is the significance of the technical contribution. The analysis of gradient estimators (Lemma 1, 2 and 4) is almost identical to Lemma 1 and 3 from Liu et al. (2018), with smoothness replaced by generalized smoothness assumption. Consequently, I do not see substantial technical novelty as claimed after Lemma 3, though it is understandable that leading lemmas might not be novel. I also urge the authors to add relevant citations, such as counterparts of lemmas that hold for $L$-smooth or first-order strategies, for better reference.

The writing of the manuscript should be carefully checked, as the current version contains much ambiguity and is error-prone. See the Questions section for details.

Reference:

Sijia Liu, Bhavya Kailkhura, Pin-Yu Chen, Paishun Ting, Shiyu Chang, and Lisa Amini. Zeroth-order stochastic variance reduction for nonconvex optimization. In Advances in Neural Information Processing Systems 31, 2018.

**Questions:**

**Typos and minor suggestions:**

Line 21: "decency" -> "dependency", "complexes" -> "complexities".

Line 101: "converge analysis" -> "convergence analysis"

Line 141: missing relevant citations for $(L_0,L_1)$-smoothness.

Line 168: it is not typical to use $f(x)-f^\star \leq \epsilon$ as criterion for $\epsilon$-stationarity, as the objective is non-convex.

Line 176 (Section 3.1): missing relevant citations for gradient estimators.

Line 181: missing explanation for parameter $\mu$.

Line 204: please capitalize "Assumption", "Lemma" and "Theorem".

Line 229: see the Weaknesses section, I assume it not difficult to derive these lemmas.

Line 238: "$\mathbf{e}\_{l}$" ->  "$\mathbf{e}\_{\ell}$".

Line 265: "introduce" -> "introduces".

Line 266: "proposed" -> "proposes".

Line 278 (Table 3): you use $c$ to refer to constant in $\eta_k$ of ZONSPIDER, while in other places you use $c_2$.

Line 296: "initialize point $x_0$" -> "initial point $x_0$".

Line 302: should the calculation of $v_0$ be different in Option I and II?

Line 307 & 308: "defied" -> "defined".

Line 307-312: the current Algorithm 1 only considers general expectation case.

Line 334: "$\Vert v_k-\nabla f(x_k)\Vert$" -> "$\Vert v_k-\hat{\nabla} f(x_k)\Vert$".

Line 335: "Sipder" -> "SPIDER".

Line 338: is there a missing $\sum_{i=1}^b$?

Line 341: "$k=q-1$" -> "$k=\hat{k}+q-1$".

Line 343: "$l=\hat{k}$" -> "$k=\hat{k}$", and should the indices in LHS be $\hat{k}+q-1$ rather than $k$?

Line 356 (Lemma 7): not consistent with Lemma E.2 w.r.t. power of $d, b$ and constants.

Line 375: "itration" -> "iterations".

Line 378: "deceacse" -> "decrease".

Line 392: "in, since $f(x)$" should be deleted.

Line 598: duplicate reference of Ji et al. (2019).

Line 1102 (last step of derivation): should be "=" due to arrangement.

Line 1111: if this lemma holds for any $b>0$, you should explicitly write it out.

Line 1117: you are analyzing the case of finite sum, however, "$\mathbb{E}[\hat{\nabla}f(x_k;\xi)]=\hat{\nabla}f(x_k)$" is for the case of general expectation.

Line 1122 (first step of derivation): should be "=" due to expansion of $v_{k+1}$.



**Questions:**

* I notice that SPIDER has provided ZO variants in the arXiv version (Fang et al., 2018). Should you consider adding it to Table 1 for comparison?
* I recommend that the authors restate the main challenges and technical contributions. My general understanding is that the additional sum of gradient norms in Lemma 7 makes it hard to bound $\mathbb{E}[\Vert v_k-\hat{\nabla}f(x_k)\Vert]$ by merely setting the parameters $c_2$ and $\mu$.
* It is somewhat surprising to see ZO methods outperform FO methods in the experiment section. In previous work, it is usually observed that ZO methods require more gradient queries than FO methods (Gautam et al., 2024). Can you provide a possible explanation for this situation or clarify the differences?


References:

Cong Fang, Chris Junchi Li, Zhouchen Lin, Tong Zhang. SPIDER: Near-Optimal Non-Convex Optimization via Stochastic Path Integrated Differential Estimator. ArXiv e-prints, arXiv:1807.01695, 2018.

Tanmay Gautam, Youngsuk Park, Hao Zhou, Parameswaran Raman, Wooseok Ha. Variance-reduced Zeroth-Order Methods for Fine-Tuning Language Models. In International Conference on Learning Representations, 2024.

---

> ### Author Response · Authors · 2024-12-03
>
> Thank you for your valuable feedback. We will incorporate your suggestions to further improve our paper.

---

### Meta-Review · Area_Chair_cuLF · 2024-12-09

**Metareview:**

This paper studies zeroth-order nonconvex optimization problems under the generalized $(L_0,L_1)$-smoothness condition. All reviewers identified significant weaknesses in the current paper, leading to a unanimous rejection. First, the novelty is limited—while the $(L_0,L_1)$-smoothness condition has not been previously studied in the context of zeroth-order optimization, the techniques used are directly taken from existing work on first-order optimization for $(L_0,L_1)$-smooth functions, with key lemmas being identical to prior studies. Additionally, the paper is poorly written, with numerous typos and unfinished sentences, which not only hinder understanding and readability but also raise concerns about technical correctness. Finally, the paper fails to discuss many classical results on zeroth-order methods in the existing literature. I encourage the authors to carefully consider these reviewer comments and rigorously improve the paper in a future version.

**Additional Comments On Reviewer Discussion:**

There is no detailed response submitted to address the reviewers' questions. No review has been changed.

---

### Decision · Program_Chairs · 2025-01-22

Reject